# Spider2-V: How Far Are Multimodal Agents From Automating Data Science and Engineering Workflows?

**Ruisheng Cao**[*12]  **Fangyu Lei**[1]  **Haoyuan Wu**[1]  **Jixuan Chen**[1]  **Yeqiao Fu**[1]  **Hongcheng Gao**[1]

**Xinzhuang Xiong**[1]  **Hanchong Zhang**[2]  **Yuchen Mao**[1]  **Wenjing Hu**[1]  **Tianbao Xie**[1]  **Hongshen Xu**[2]

**Danyang Zhang**[12]  **Sida Wang**  **Ruoxi Sun**[3]  **Pengcheng Yin**[4]  **Caiming Xiong**[5]  **Ansong Ni**[6]

**Qian Liu**[7]  **Victor Zhong**[8]  **Lu Chen**[2]  **Kai Yu**[2]  **Tao Yu**[1]

[1] The University of Hong Kong   [2] Shanghai Jiao Tong University

[3] Google Cloud AI Research   [4] Google DeepMind   [5] Salesforce Research

[6] Yale University   [7] Sea AI Lab   [8] University of Waterloo

ruishengcao@gmail.com   tyu@cs.hku.hk

## Abstract

Data science and engineering workflows often span multiple stages, from warehousing to orchestration, using tools like `BigQuery`, `dbt`, and `Airbyte`. As vision language models (VLMs) advance in multimodal understanding and code generation, VLM-based agents could potentially automate these workflows by generating SQL queries, Python code, and GUI operations. This automation can improve the productivity of experts while democratizing access to large-scale data analysis. In this paper, we introduce Spider2-V, the first multimodal agent benchmark focusing on professional data science and engineering workflows, featuring 494 real-world tasks in authentic computer environments and incorporating 20 enterprise-level professional applications. These tasks, derived from real-world use cases, evaluate the ability of a multimodal agent to perform data-related tasks by writing code and managing the GUI in enterprise data software systems. To balance realistic simulation with evaluation simplicity, we devote significant effort to developing automatic configurations for task setup and carefully crafting evaluation metrics for each task. Furthermore, we supplement multimodal agents with comprehensive documents of these enterprise data software systems. Our empirical evaluation reveals that existing state-of-the-art LLM/VLM-based agents do not reliably automate full data workflows (14.0% success). Even with step-by-step guidance, these agents still underperform in tasks that require fine-grained, knowledge-intensive GUI actions (16.2%) and involve remote cloud-hosted workspaces (10.6%). We hope that Spider2-V paves the way for autonomous multimodal agents to transform the automation of data science and engineering workflow. Our code and data are available at `https://spider2-v.github.io`.

## 1   Introduction

Data science and engineering pipelines usually rely on professional data software systems such as `BigQuery`, `dbt`, and `Airbyte` to acquire, process, and orchestrate large-scale data. Utilizing these enterprise systems involves writing SQL and Python code, as well as frequent and repetitive graphical user interface (GUI) controls, which can be complex even for experienced data scientists and engineers. With rapid advances in large language models (LLMs) and vision language models (VLMs), LLM/VLM-based autonomous agents have the potential to automate these work-

---

*   Work done while interning at the University of Hong Kong.

38th Conference on Neural Information Processing Systems (NeurIPS 2024) Track on Datasets and Benchmarks.

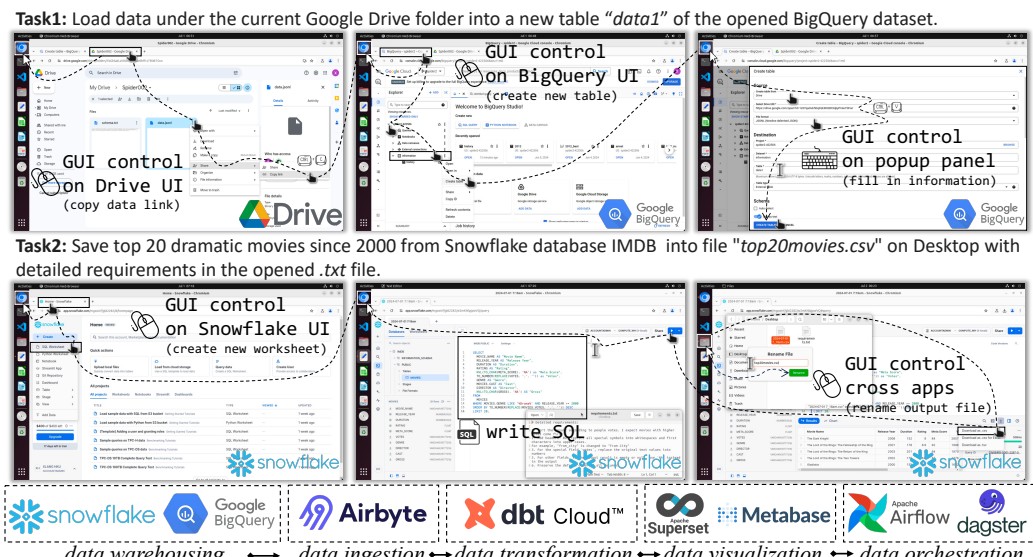

**Task1:** Load data under the current Google Drive folder into a new table *"data1"* of the opened BigQuery dataset.

**Task2:** Save top 20 dramatic movies since 2000 from Snowflake database IMDB into file "*top20movies.csv*" on Desktop with detailed requirements in the opened *.txt* file.

*data warehousing* ⟷ *data ingestion* ⟷ *data transformation* ⟷ *data visualization* ⟷ *data orchestration*

Figure 1: Spider2-V is a multimodal agent benchmark spanning across complete data science and engineering workflows (*e.g.*, two task examples in the Figure above). It involves various professional enterprise-level applications and includes intensive GUI controls apart from code writing throughout the real-time multi-turn interaction with an executable computer environment.

flows [40, 35], enhancing productivity for data scientists and engineers [41, 18] while democratizing access to large-scale data [17, 43].

Previous studies on data agents focused mainly on daily life data processing and analysis by generating code or API calls [45, 11, 4], neglecting other crucial stages of data science and engineering (*e.g.,* data ingestion and integration) using enterprise applications (*e.g.,* Snowflake, Airflow, and Dagster). Additionally, to complete data workflows, data scientists and engineers often need to navigate multiple professional data systems, combining code writing with intensive GUI controls, such as navigating web pages and clicking buttons [5, 48]. However, there is currently no benchmark that integrates both code generation and GUI controls for professional data science and engineering.

To address this gap, we propose Spider2-V, the first multimodal agent benchmark covering the entire data science and engineering workflow, involving 494 real-world tasks in a real-time executable computer environment and 20 professional enterprise data software. Spider2-V aims to evaluate a multimodal agent's ability to perform professional data-related tasks by writing code and managing the GUI in enterprise data software systems, including data warehousing (*e.g.,* BigQuery), data ingestion and integration (*e.g.,* Airbyte), data transformation (*e.g.,* dbt), data analysis and visualization (*e.g.,* Superset), and data orchestration (*e.g.,* Dagster). These tasks are derived from real-world practices, such as official tutorials on professional applications and open-source data engineering projects (with two task examples presented in Figure 1). We also supplement retrieval-augmented agents with official documentation and tutorials of these software systems to assess their capability to generalize and learn from these resources.

Each task in Spider2-V is defined within an executable computer environment based on OS-WORLD [37], which allows multimodal agents to simulate human actions (*e.g.*, typing code or clicking buttons) in a realistic setting. Specifically, a multimodal agent can observe real-time image-style screenshots and text-style accessibility tree of professional data applications in the current workflow and execute its predicted actions in dynamic multi-round interaction with the computer. This environment is connected to the real-world Internet, allowing the inclusion of professional software requiring authentic user accounts (*e.g.,* Snowflake). To ensure reproducible and reliable experiments with this enterprise data software, 10 authors with computer science backgrounds developed 170 automatic task setup configurations and 151 customized evaluation metrics in total.

We experiment with state-of-the-art LLMs and VLMs including closed-source ones GPT-4 series [23], Gemini-Pro-1.5 [28], Claude-3-Opus [2], QWen-Max [3] and open-source representatives Mixtral-

8x7B [13] and Llama-3-70B [22]. Performances reveal that even the top-tier VLM (GPT-4V [1]) achieves only $14.0\%$ success rate. In the most challenging subset, with action steps exceeding $15$, the performance drops to $1.2\%$. And for those open-source LLMs, the success rate is less than $2\%$. This indicates that existing LLMs or VLMs are still far away from achieving full data workflow automation. Even provided with an oracle step-by-step plan, the overall performance only increases to $16.2\%$. This observation uncovers the poor capability of action grounding (*e.g.*, identifying the precise coordinates of elements in the current focused application window) for multimodal agents. Furthermore, extensive analysis (§ 4.3) on Spider2-V demonstrate that these strategies remarkably promote the final performance, which include enhancing the alignment between different observation modalities, introducing feedback on action execution, integrating retrieved document context and enlarging the history trajectory length. These findings lay the groundwork for developing practical multimodal agents that can revolutionize the automation of data science and engineering workflows.

## 2    Executable Computer Environment of Spider2-V

In this section, we introduce the real-time executable computer environment of Spider2-V, which is built upon virtual machines (VMs) and adapted from OSWORLD [37].

### 2.1    Task Definition

Generally, an autonomous data agent is modeled as a partially observable Markov decision process (POMDP). Given the current observation $o_t \in \mathcal{O}$ which includes a natural language instruction and a screenshot, accessibility tree (`a11ytree`), or their combination, an agent generates an executable action $a_t \in \mathcal{A}$. This action can be clicking on a certain pixel of the screen (`CLICK(560, 200)`), or writing code through keyboard (`TYPE("ls -lh")`). The execution of $a_t$ results in a new state $s_{t+1} \in \mathcal{S}$ (*e.g.*, the updated computer state) and a new partial observation $o_{t+1} \in \mathcal{O}$. The `a11ytree` is a text-style representation of the desktop environment, which describes the status, position, and text content of each element (e.g., windows, buttons, and input boxes). The interaction loop repeats until an action that marks termination (`DONE` or `FAIL`) is generated or the agent reaches the max number of steps. See App. D for more details about the observation space and action space.

### 2.2    Environment Setup

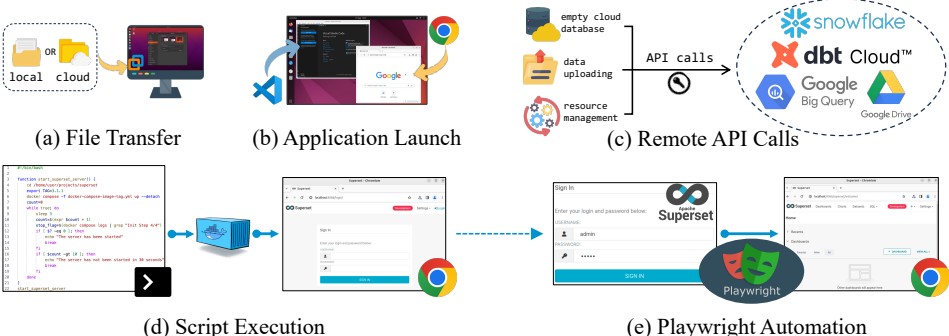

(a) File Transfer       (b) Application Launch       (c) Remote API Calls

(d) Script Execution                (e) Playwright Automation

Figure 2: Five common operations to reset the initial environment.

To ensure that an agent starts from a consistent initial state, we invoke a series of function calls from a pre-stored virtual machine (VM) snapshot to reset the environment. These function calls vary among tasks, resulting in 170 initial states. And we summarize 5 universal categories (see Figure 2), namely: 1) *File Transfer*: transfer files or project archives (either from local or cloud storage) into the VM; 2) *Application Launch*: open software on the desktop, *e.g.,* Visual Studio Code and Chromium; 3) *Remote API Calls*: invoke tool-specific API calls for professional applications, especially those requiring authentic user accounts, to reset and configure cloud workspaces; 4) *Script Execution*: execute a shell script in VM to set up the initial state, *e.g.,* run a Docker container to start a localhost webserver for `Superset`; 5) *Playwright Automation*: run web browser simulation with Playwright, *e.g.,* sign into an account or click a specific button and redirect to the target web page.

## 2.3 Task-specific Evaluation

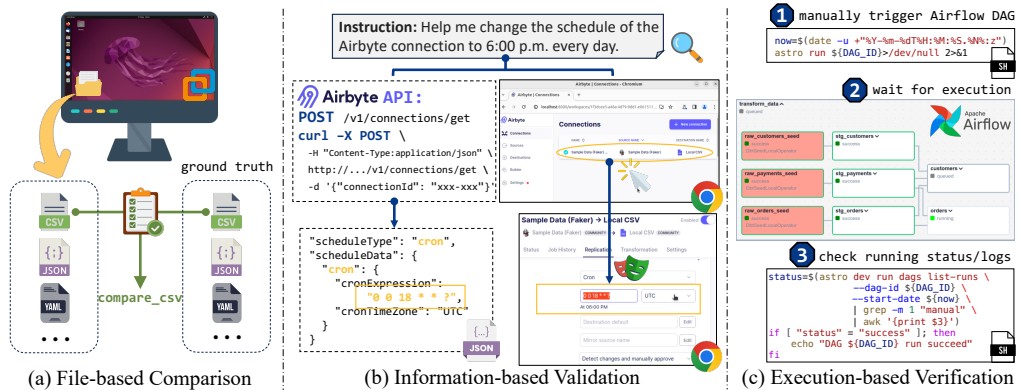

Figure 3: Three generic methods for task evaluation.

After the interaction terminates, we only have access to the open-ended resulting state of the computer. Thus, to measure whether the goal of each task is accomplished, we write task-specific functions to retrieve the desired result from the open-ended resulting state and return the success flag ($0/1$). All evaluation methods (in total, $151$ evaluation scripts across the entire benchmark) can be classified into 3 generic categories, also shown in Figure 3:

a) *File-based comparison*: this method finds and copies the target files from VM to the host, and resorts to file-type based metrics (e.g., `.json`, `.csv`, etc.) to compare the specified aspect of the generated file with ground truth. Sometimes, the ground truth may be updated over time. In this case, we will fetch the latest labels from the Internet during evaluation.

b) *Information-based validation*: this scheme is usually utilized to extract and check desired information from the computer. For example, in Figure 3(b), we want to confirm whether the time schedule of the data transportation is correctly configured in `Airbyte`. We can invoke `Airbyte` APIs to retrieve, or Chromium Playwright to locate the target value.

c) *Execution-based verification*: to verify whether an expected goal is achieved, we may also need to first execute a complicated Shell script in the final VM. For example, in Figure 3(c), we manually trigger the target `Airflow` DAG [2] and check the eventual status through running logs.

## 3 Benchmark Construction

In this section, we introduce the general annotation pipeline, document warehouse construction, and dataset statistics for Spider2-V. For concrete examples, refer to App. H.

### 3.1 Annotation Pipeline

To construct tasks in different categories, we find that official tutorials of enterprise applications serve as an excellent starting point. The 6-step annotation pipeline is illustrated in Figure 4(a), and we elaborate it with a concrete and real example "*Orchestrate dbt Core jobs with Airflow and Cosmos*" [3]:

1) **Collect tutorials:** firstly, we find tutorials from official websites for each professional tool in Figure 5. In total, 10 annotators collected 217 source URLs. Note that these tutorials may utilize other professional software, e.g., MySQL. All involved professional tools are listed in App. B.

2) **Learn tutorials:** the annotator selects one tutorial, learns and realizes it in the VM. After that, they can summarize key knowledge points from this tutorial. For example, in Figure 4(b), five key steps in integrating a `dbt` project into an `Airflow` task are extracted.

---

[2] A DAG in `Airflow` is defined as a collection of tasks to run, and DAG_ID is used to uniquely identify it.

[3] The selected `Airflow` tutorial URL: `https://www.astronomer.io/docs/learn/airflow-dbt`

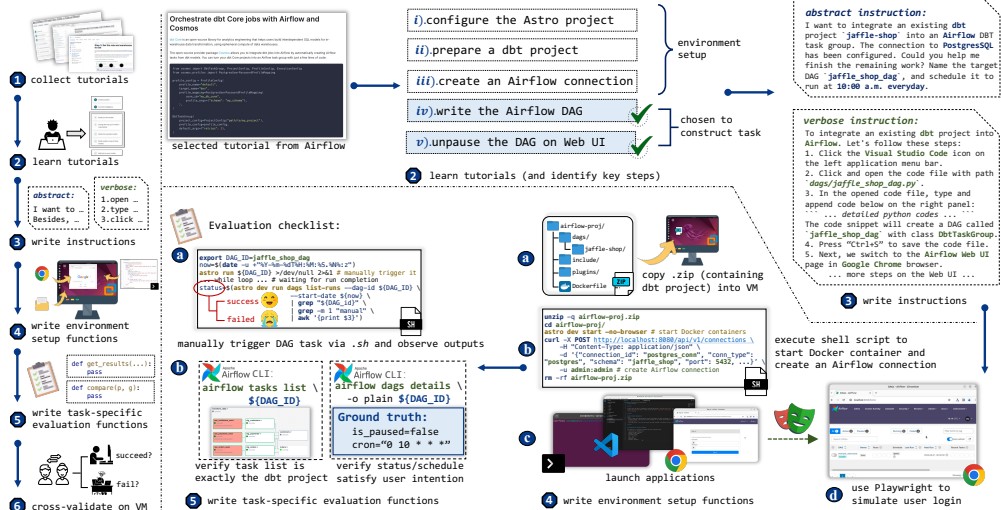

Figure 4: The annotation pipeline of one selected demonstration from the `Airflow` tutorial: *Orchestrate dbt Core jobs with Airflow and Cosmos*. On average, annotating each task costs roughly 4 hours.

3) **Write instructions:** since the chosen tutorial is extremely complicated, the annotator can select a few key points to construct the task instruction. In Figure 4, we only select key steps *iv)* and *v)* to write two versions of instructions, *abstract* and *verbose*, indicating different levels of proficiency. Note that, to avoid potential data contamination and make the task more realistic, we ask the annotator to introduce at least two modifications to the raw tutorial. In this example, we a) replace the original "`my_simple_dbt_project`" into an open-source `dbt` project called "`jaffle-shop`" [4], and b) add one extra requirement on the time schedule (10:00 a.m. daily).

4) **Write environment setup functions:** the next step is to write initialization functions using operations defined in § 2.2. In the example above, we need to: a) Upload an unfinished `Airflow` project into the VM. b) Execute a Shell script to launch the web server (via Docker containers) for `Airflow` under the project folder. c) Open all relevant applications on the desktop to simulate real user scenarios. d) Use Playwright to auto-login to the default `Airflow` account.

5) **Write task-specific evaluation functions:** In this step, annotators are required to programmatically obtain results from the open-ended states of VM and assess whether the task is completed using methods in § 2.3. In this example, the evaluator contains: a) manually run the target `Airflow` DAG and verify the final status is "success"; b) using `Airflow` CLIs to retrieve details of the target `Airflow` DAG, and compare `dbt` sub-tasks, status and schedule with ground truth.

6) **Cross-validate on VM:** to ensure correctness, we go through strict cross-validation. Each annotated task is sent to two other annotators to check: a) whether the chosen task reflects a real-world use case; b) whether verbose instruction accurately fulfills the task and its requirements in the abstract instruction; c) whether the environment can be reset to the same state in different trials; d) whether the evaluation is robust when we exactly follow the verbose instruction or modify some inconsequential steps; e) whether the evaluation score is 0 if we deliberately make some mistakes (red-teaming). The task is preserved only if it withstands all these tests.

## 3.2 Document Warehouse

Even senior data scientists query official documentation of professional applications when completing a complicated data engineering task. To compensate for the deficiencies of the data agents in utilizing enterprise professional software (e.g., unaware of coding specifications or APIs), we build a document warehouse for Spider2-V. Concretely, we recursively crawl the web pages from the root websites of the professional applications in Figure 5. After pre-processing through heuristics (refer to App. C), raw HTML web pages are convert into 3 different formats for retrieval, namely a) pure text, b) markdown, and 3) simplified HTML. Eventually, we obtain 11, 231 documents in total.

---

[4]URL of open-source `dbt` project "`jaffle-shop`": `https://github.com/dbt-labs/jaffle-shop`

## 3.3 Dataset Statistics

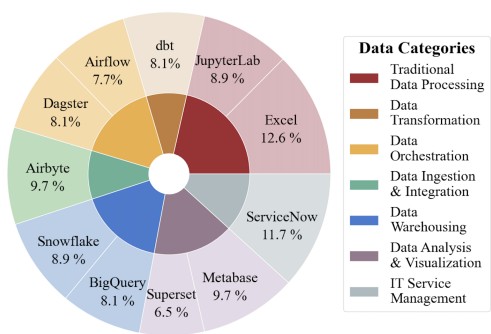

Figure 5: Task categories with professional tools.

Table 1: Statistics of Spider2-V.

| Statistics | Number |
|---|---|
| **Total Tasks** | **494 (100%)** |
| - Pure CLI | 28 (5.7%) |
| - Pure GUI | 186 (37.7%) |
| - CLI + GUI | 280 (56.7%) |
| - w. Authentic User Account | 170 (34.4%) |
| - w/o. Authentic User Account | 324 (65.6%) |
| **Level (Action Steps)** | |
| - Easy ($\leq 5$) | 98 (19.8%) |
| - Medium ($6 \sim 15$) | 310 (62.8%) |
| - Hard ($> 15$) | 86 (17.4%) |
| Avg. Action Steps | 4.0 / 9.6 / 22.0 |
| Avg. Length of Abstract Instructions | 37.1 |
| Avg. Length of Verbose Instructions | 191.5 |
| Avg. Number of Used Apps Per Task | 2.5 |

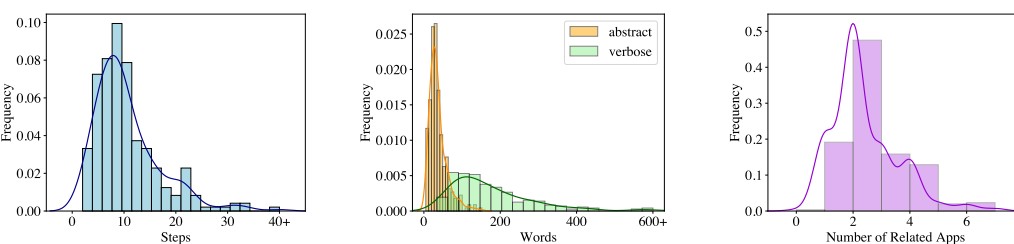

Figure 6: Distribution of action steps, instruction length, and related applications per task.

**Tasks** We classify all $494$ tasks in Spider2-V into 7 categories and 11 software sub-categories with main statistics in Figure 5 and Table 1. Specifically, most (280 tasks, $56.7\%$) involve CLI and GUI operations. And $34\%$ examples request registering authentic software accounts. Since each task is associated with a detailed, step-by-step tutorial (verbose instruction), the entire task set can be categorized into three distinct levels based on the number of actions in these instructions. The proportion of easy, medium, and hard tasks is approximately $1 : 2 : 1$. According to the rightmost distribution depicted in Figure 6, most tasks necessitate the coordinated utilization of multiple professional applications, thereby establishing Spider2-V as a particularly challenging benchmark.

**Comparison with existing benchmarks** In Table 2, we compare Spider2-V with other agent benchmarks. Spider2-V incorporates generic computer control commands into the field of data science and engineering and is distinguished by these salient features: 1) a real-time executable environment. Instead of providing static input-output pairs, Spider2-V is equipped with a dynamic computer desktop such that agents can proactively explore it; 2) multiple enterprise software. We integrate 20 professional applications into the benchmark, which include not only tools installed on local hosts but also cloud-based enterprise services; 3) intensive GUI operations. Unlike traditional coding or data science domains, experienced data scientists frequently manipulate the UIs of those professional software to simplify the data workflow (*e.g.*, enabling a specific function on the UI page or visualizing the graph view of data inputs). In summary, Spider2-V focuses on the use of professional enterprise software with visual interface in an interactive computer environment.

## 4 Experiments and Analysis

In this section, we introduce the experiment settings, experimental results, and ablation study to assess the proficiency of current LLM or VLM based agents on Spider2-V benchmark.

Table 2: Comparison with existing agent benchmarks. Columns include the research field (Field), whether an executable environment is provided (Exec. Env.?), whether enterprise service is utilized (Ent. Serv.?), whether GUI actions are supported (GUI Support?) and some other statistics.

| Benchmark | Field | Exec. Env? | Ent. Serv.? | GUI Support? | # Apps/ Sites | # Exec.-based Eval. Func. | # Tasks |
|---|---|---|---|---|---|---|---|
| Spider [45] | Text-to-SQL | ✗ | ✗ | ✗ | 1 | 0 | 1034 |
| DS-1000 [17] | Data Science | ✗ | ✗ | ✗ | 1 | 0 | 1000 |
| Arcade [43] | Data Science | ✗ | ✗ | ✗ | 1 | 0 | 1082 |
| MLAgentBench [12] | Machine Learning | ✓ | ✗ | ✗ | 4 | 13 | 13 |
| SWE-Bench [14] | Software Engineering | ✗ | ✗ | ✗ | 12 | 1 | 2294 |
| Mind2Web [5] | Web | ✗ | ✗ | ✓ | 137 | 0 | 2000 |
| WEBLINX [21] | Web | ✗ | ✗ | ✓ | 155 | 0 | 2337 |
| WorkArena [6] | Web | ✓ | ✓ | ✓ | 1 | 7 | 29 |
| AppAgent [46] | Android | ✓ | ✗ | ✓ | 10 | 0 | 50 |
| AndroidWorld [27] | Android | ✓ | ✗ | ✓ | 20 | 6 | 116 |
| WebArena [48] | Web | ✓ | ✗ | ✓ | 5 | 5 | 812 |
| OSWorld [37] | Computer Control | ✓ | ✗ | ✓ | 9 | 134 | 369 |
| Spider2-V | Data Science & Engineering w/ Computer Control | ✓ | ✓ | ✓ | 20 | 151 | 494 |

## 4.1 Environment Settings

**Agent baselines**  The baseline method includes 3 schemes in zero-shot prompt learning: 1) Set-of-Mark (SoM, [39]): following OSWORLD [37] and VisualWebArena [16], we adopt heuristic methods to retrieve coordinates of visible elements from `a11ytree` (a text-format observation type) and draw indexed bounding box for these elements on the screenshot. We further insert these indexes into the pruned `a11ytree` to enhance the alignment between screenshot and `a11ytree`. 2) Execution Feedback (EF, [30]): we append execution feedback messages of those actions which failed to be grounded in the environment due to unexpected errors. The two techniques mentioned above are elaborated in App. D.3.1. 3) Retrieval-Augmented Generation (RAG, [8]): we leverage the task instruction as the query vector, `bge-large-en-v1.5` [36] as the embedding model, and LlamaIndex [20] framework as the retrieval to generate document context for each task example. Documents are pre-chunked into segments with maximum length 512 and tokens overlapping size 20. Top 4 segments are selected as additional context in the task prompt (detailed in App. I.3).

**LLMs and VLMs**  We experiment with state-of-the-art LLMs and VLMs, including open-source representatives such as Mixtral-8x7B [13] and Llama-3-70B [22], and closed-source ones including Qwen-Max [3], Gemini-Pro-1.5 [28], Claude-3-Opus [2] and GPT [1] families (GPT-4o and GPT-4V [5]). With respect to the two open-source LLMs and QWen-Max, we utilize pure text-format `a11ytree` as the observation type on account of their incapability of image processing. For the remaining 4 VLMs which support vision input, we use aligned text and image (that is Set-of-Mark) as the observation type in main experiments. Unless otherwise specified, we set the temperature to 0.5 and top_p to 0.9, the history trajectory window size to 3, the maximum length of `a11ytree` to 5000 tokens, and the maximum output tokens to 1500 in each turn. Heuristically, we require the agent to complete the tasks within both 15 interaction turns and one hour, which suffices for most tasks [6].

## 4.2 Main Results

In Table 3, we compare performances of different LLMs and VLMs. All results above integrate techniques of both execution feedback (EF) and retrieval-augmented generation (RAG) in § 4.1. Accordingly, we can summarize that:

1) **Existing data agents are far from satisfactory in completing real-world data science and engineering tasks.** Even state-of-the-art VLMs (GPT-4o and GPT-4V) perform terribly on

---

[5]We utilize the version `gpt-4o-2024-05-13` for GPT-4o and `gpt-4-1106-vision-preview` for GPT-4V.

[6]Although some tasks require more than 15 actions, we encourage the multimodal agent to predict multiple actions in one response in order to save the budget in the prompt design (see App. I.1.2).

Table 3: Success rates of baseline agents on Spider2-V grouped by 7 task categories (see Figure 5), namely data warehousing (*ware.*), transformation (*trans.*), ingestion (*inges.*), visualization (*visual.*), orchestration (*orche.*), traditional data processing (*proc.*), and IT service management (*manag.*). For the first three LLMs, since they do not support visual information, we only utilize the text-based `a11ytree` as the observation. For the remaining four VLMs, we adopt Set-of-Mark (see § 4.1).

| LLM / VLM | Observation | Success Rate (%) | | | | | | | |
|---|---|---|---|---|---|---|---|---|---|
| | | *ware.* | *trans.* | *inges.* | *visual.* | *orches.* | *proc.* | *serv.* | **Overall** |
| Mixtral-8x7B | | 1.2 | 0.0 | 0.0 | 0.0 | 2.6 | 0.9 | 0.0 | 0.8 |
| Llama-3-70B | a11ytree | 2.4 | 0.0 | 0.0 | 2.5 | 3.9 | 2.8 | 0.0 | 2.0 |
| Qwen-Max | | 1.2 | 0.0 | 0.0 | 0.0 | 2.6 | 0.0 | 0.0 | 0.6 |
| Claude-3-Opus | | 2.4 | 2.5 | 10.4 | 15.0 | 11.5 | 3.8 | 12.1 | 8.1 |
| Gemini-Pro-1.5 | Set-of-Mark | 3.6 | 2.5 | 14.6 | 15.0 | 10.3 | 2.8 | **19.0** | 9.1 |
| GPT-4o | | 7.2 | 7.5 | **24.0** | 14.1 | **19.8** | **10.1** | 13.8 | 13.8 |
| GPT-4V | | **10.8** | **10.0** | 12.0 | **25.0** | 18.4 | 8.5 | 12.1 | **14.0** |

Spider2-V, achieving at best $14.0\%$ overall success rate. As for their strongest competitors, Gemini-Pro-1.5 [28] and Claude-3-Opus [2], they attain worse performances, even less than $10\%$ percents. There is still ample room for improvement in future work.

2) **Closed-source models are much more superior than open-source ones**. For those open-source LLMs, the success rate is less than $2\%$, with some categories approaching zero. On one hand, it can be attributed to the fact that closed-source VLMs are pre-trained and fine-tuned on data of higher quality. On the other hand, closed-source VLMs support inputs with longer contexts and integrate both vision and text modalities (further analyzed in § 4.3).

3) **Performances of data agents exhibit high variance, especially in categories "*data ingestion*" and "*data visualization*".** The majority of these two partitions are pure GUI tasks, which means agents mostly interact with the environment through time-dependent GUI operations. However, a minor error in one intermediate step can be amplified, resulting in the entire sequence of actions being wasted. Through error analysis on trajectories, we discover that once agents mispredict the coordinates of the correct button, they will open the wrong window and become trapped in the incorrect area, unable to return.

4) **Across 7 data categories, the partitions "*data warehousing*" and "*traditional data processing*" are challenging, both less than $10\%$ success rates.** The reasons for this observation are two-fold: a) *data warehousing* tasks mostly involve authentic user accounts (*e.g.*, `BigQuery` and `Snowflake`). Compared to other tasks which can be accomplished in a local host, these dynamic real-world scenarios incur extra burden on data agents, such as network connection delay and pop-up windows. Multimodal agents need to deal with these unexpected situations in real-time interaction with the computer. b) As for *traditional data processing*, the bottleneck is that spreadsheets in Excel contain many cells, and it is particularly difficult for data agents to accurately locate the coordinates of cells. For example, applying the same math formula to the entire column requests multimodal agents to firstly pinpoint the right corner of a specific cell, wait for the mouse to become a cross, press and drag the mouse towards the target cell. This series of actions requires precise and fine-grained GUI controls which are difficult to implement.

### 4.3 Analysis

In this section, we delve into different factors which influence the eventual success rates, and analyze the underlying logics. The following analyses are based on our agent baseline with VLM GPT-4o unless otherwise specified. Firstly, we split the overall results into different subsets in Table 4.

1) **Tasks with more inherent action steps are more difficult.** Each task is associated with one verbose task instruction which gives a step-by-step guidance on how to complete it. We count the number of actions in the verbose instruction and split the entire task set into 3 difficulty levels: $\leq 5$ steps (Easy), $5 \sim 15$ steps (Medium), and $> 15$ steps (Hard). Not surprisingly, as the number of intrinsic action steps increases, the average performance decreases significantly. And for tasks with steps more than 15, existing VLM-based data agents can hardly accomplish the goal.

Table 4: Success rate of GPT-4o with agent baseline SoM+EF+RAG across different partitions.

| Task Splits | Ratio (%) | SR (%) |
|---|---|---|
| Easy | 19.8 | **38.8** |
| Medium | 62.8 | 9.7 |
| Hard | 17.4 | 1.2 |
| w/o account | 66.0 | **15.6** |
| w/ account | 34.0 | 10.6 |
| CLI | 5.7 | 7.1 |
| GUI | 37.7 | **20.1** |
| CIL+GUI | 56.7 | 10.6 |
| Abstract | 50 | 11.3 |
| Verbose | 50 | **16.2** |

Table 5: Ablation study on action space, observation types and 3 methods in § 4.1 on task subset.

| Action Space | Observation Types | SR (%) |
|---|---|---|
| JSON dict | screenshot | 4.2 |
| pyautogui | | 4.2 |
| JSON dict | a11ytree | 10.5 |
| pyautogui | | **12.6** |
| | screenshot+a11ytree | 11.4 |
| | w/ Set-of-Mark | 15.6 |
| pyautogui | w/ exec. feedback | 13.6 |
| | w/ retrieval aug. | 14.4 |
| | w/ all tricks | **16.3** |

2) **Tasks involving authentic user accounts are much more challenging.** One salient feature of Spider2-V is the integration of professional applications that require authentic user accounts. We also split the entire task set accordingly (w/o or w/ account). Notably, data agents struggle to complete tasks involving authentic user accounts (10.6% success rate). These tasks deal with real-world scenarios and incorporate cloud-hosted enterprise services. Compared with Web servers which are launched locally in the VM (*e.g.*, from Docker containers), the cloud Web UIs 1) generally integrate more comprehensive functionalities or options in their menu panel, and 2) potentially suffer from emergency situation, such as extended network response delay due to bandwidth limitation or server overload. We conjecture these two causes collectively contribute to the inferior performances.

3) **Incorporating GUI operations typically lead to improved performances.** We split the task set by interfaces. If the task can be completed with pure CLIs (e.g., code editor or bash terminal), we classify it as CLI. If the task only requires the agent to manipulate the GUI (usually on the Web page), we classify it into GUI. For the remaining cases (CLI+GUI), an agent must write code or scripts, and control the UI screen. We observe that pure GUI tasks are much easier than CLI tasks. This conclusion can be explained by the following two reasons: 1) GUIs of professional applications are designed to simplify the original coding task. Clicking buttons or typing values on UIs can avoid handling the rigorous and complex coding specification. 2) Both observation types, namely the screenshot and a11ytree, are naturally proposed for GUI tasks. For pure CLI tasks, data agents must perform extra actions to locate and switch to the target panel before writing code.

4) **Providing a step-by-step guideline in task instructions results in remarkable performance gains.** The key difference between abstract and verbose instructions (the third step in § 3.1) is whether a detailed step-by-step guidance is offered. With such stepwise oracle tutorials, data agents do not need to reason and plan, thus dramatically simplifying the original task. And the 4.8 points improvement in Table 4 consolidates this hypothesis. Nevertheless, the low success rate with verbose instructions (16.2%) indicates that current VLMs still yield unsatisfactory results when purely grounding actions in real-world contexts. And significant potential remains for further enhancement.

In Table 5, we analyze the influence of different combinations of action space, observation types, and the 3 techniques described § 4.1. The findings include: 1) **Regarding action space, pyautogui code slightly outperforms self-customized JSON dict (12.6% v.s. 10.5%).** This can be attributed to the advantage that agents can also generate functional Python code like file traversal apart from the limited GUI control operations using the first action space. And it improves the efficiency of action grounding. 2) **As for observation types, single screenshot leads to very low performances (4.2%) on account of the agent's failure in pinpointing concrete elements.** When inserting a11ytree into the observation which contains precise coordinates, the agent capability of locating target pixels is remarkably promoted. 3) **All 3 methods we integrate into the agent baseline (namely SoM, EF and RAG) will boost eventual performances.** It is interesting that if we do not adopt Set-of-Mark (that is, enhancing the alignment between two modalities of observations), the result of screenshot+a11ytree is even worse than that using pure a11ytree. This emphasizes the significance of modal alignment when handling state observations.

**A moderate temperature and longer history window size improve performances.** In Figure 7, we investigate the influences of two hyper-parameters on a task subset: 1) The top-ranked performance is achieved with sampling temperature $0.5$. 2) With the history window size enlarges, from $0$ (no history, only the current observation) to $3$, the performance in-

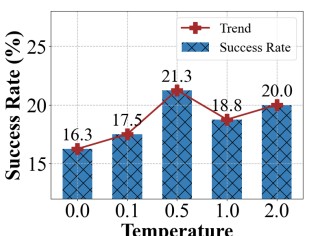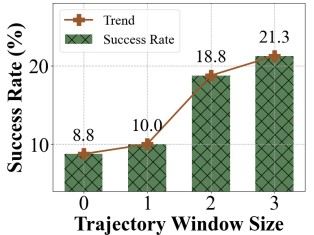

Figure 7: Ablation study on hyper-parameters.

creases stably. However, due to constraints on input length and considerations of cost-effectiveness, we are unable to extend the history trajectories any further. This also points out that the interaction efficiency is a serious issue and promising research direction.

## 5 Related Work

**Benchmarks for data science and engineering** In the field of data science and engineering, several recent works propose novel benchmarks to evaluate the capabilities of LLM agents in manipulating Excel spreadsheets [18, 4], common data science libraries (*e.g.*, `SQL` and `pandas`) [45, 17, 11, 43], machine learning [12] or software engineering [18] projects. They are usually confined to a single stage within the entire data pipeline, predominantly data processing and analysis, thus overlooking other stages such as data warehousing and orchestration from a broader perspective. Besides, like other coding-related datasets [41, 32, 44], they merely focus on the command line interface, neglecting the fact that enterprise software usually has rich graphical user interfaces (GUIs). And data scientists often combine code programming with intensive GUI operations to fulfill a data workflow. To this end, Spider2-V is proposed as the first-of-its-kind multimodal agent benchmark in the field of data science and engineering, which covers the entire data workflow and integrates visual interfaces.

**Benchmarks for multimodal agents** Existing works on GUI interaction mainly encompass web navigation [29, 19, 42, 5, 16], mobile device [46, 47, 26, 27, 33], and computer desktop [37, 35, 7, 15]. One trend of recent advanced benchmarks is to provide an executable simulation environment. Multi-modal agents can explore and interact with this platform through keyboard, mouse, gesture and touch screen actions in a more realistic and complex scenario. However, previous literature mostly focuses on daily life applications (*e.g.*, Web browser and calendar) [38, 25] or workflows of non-specialized business tasks [34]. Few works [6, 37, 34] investigate the capability of multimodal agents to manipulate enterprise-level software. GUIs of professional applications often contain abundant domain-specific terminologies (*e.g.*, "*materialization*" in `Dagster`), which requires multimodal agents to understand the specialized knowledge. Spider2-V incorporates $20$ professional tools into a real-time computer environment to test the proficiency of agents in data science and engineering. Furthermore, we supplement a large volume of documents for retrieval to compensate for deficiencies of agents in domain knowledge.

## 6 Conclusion

In this work, we propose Spider2-V, the first data science and engineering benchmark which integrates enterprise professional applications and supports intensive GUI operations besides code writing across the full data pipeline. It contains $494$ tasks, involves $20$ professional tools, and provides a real-time executable computer environment. The most advanced VLM (GPT-4V) still performs poorly on Spider2-V (achieving $14.0\%$ success rate), rendering it a very challenging benchmark. Although current multimodal agents are still far from automating data workflows, Spider2-V presents an easily accessible benchmark and lays the foundation for future research.

## Acknowledgments and Disclosure of Funding

We would like to thank Yiheng Xu, Hongjin Su, Xiaochuan Li, and Toh Jing Hua for their helpful assistance and feedback on this work.

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

## A  Relevant URLs

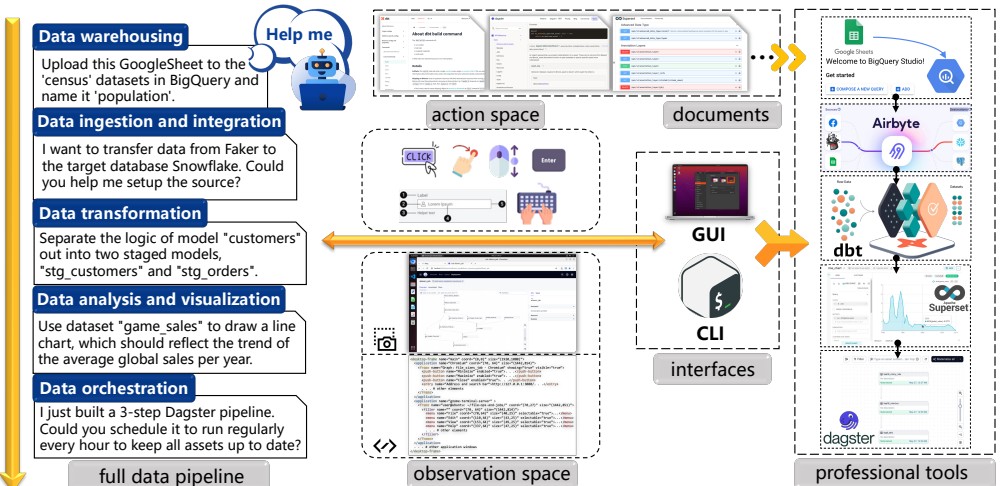

Figure 8: Overview of Spider2-V, which includes task examples across the full data pipeline, an executable computer environment, and a document warehouse for agent retrieval.

**Github Repository**  The task examples, environment, documents, code and experiments are publicly available in Github repository `https://github.com/xlang-ai/Spider2-V` under **Apache-2.0 license**. Both the environment and task examples will be maintained by the authors continuously.

Concretely, the environment code is adapted from previous work OSWORLD [37], which is released under Apache-2.0 license. A non-exhaustive list of artifacts (or task examples) used in Spider2-V includes: 1) SheetCopilot [18] which is released under GPL-3.0 license, 2) WorkArena [6] which is distributed under Apache-2.0 license, and 3) official tutorials or guides on professional applications (e.g., `dbt`, `Airflow`, `Dagster`, `Superset`, etc.). These tutorials are free to use and publicly available. For those enterprise applications which require real accounts, namely `BigQuery`, `Snowflake`, `dbt-cloud` and `ServiceNow`, we only exploit their sandbox functions or free-trials without introducing any extra cost or privacy issues.

**Project Website**  We also build a project website `https://spider2-v.github.io/` based on Nerfies [24] template which is free-to-use and licensed under a Creative Commons Attribution-ShareAlike 4.0 International License. On this website, we provide a high-level overview of Spider2-V, the leaderboard of the benchmark, and more concrete dynamic task demonstrations on a separate explorer Web page `https://spider2-v.github.io/explorer.html`.

The authors declare that the benchmark collection and usage strictly obey the aforementioned licenses.

## B  Checklist of All Professional Software in Spider2-V

In Table 6, we list all professional tools incorporated in the Spider2-V benchmark, as well as their categories and descriptions.

## C  Details of Document Warehouse

### C.1  Document Websites for Professional Tools

Table 8 lists the official documentation websites corresponding to different software. We crawled only the English documentation from each official website and selected documents matching the version installed in our testing environment for download. We used `HTTrack` [7], a free and easy-to-use offline

---

[7]`https://www.httrack.com/`

Table 6: Summary of all applications in Spider2-V (label ♡ means a real account is needed).

| Category | Software | Description |
|---|---|---|
| Data Warehousing | BigQuery♡ | Fully-managed enterprise data warehouse service offered by Google Cloud Platform (GCP). It enables rapid processing and analysis of large datasets using SQL-like queries. |
| | Snowflake♡ | Cloud-based data warehousing and analytics platform for large-scale data storage and processing, providing services to load, store, query, and analyze datasets at scale. |
| | MySQL | High-performance and scalable relational database management system (RDBMS) that is widely used and suited for fast data retrieval. |
| | PostgreSQL | RDBMS to store and manage large amounts of data with extensive additional features. |
| | DuckDB | Self-contained, serverless RDBMS with column-store architecture for fast analytical queries. |
| | SQLite | Another lightweight and serverless RDBMS that optimizes queries or transactions on individual rows. |
| Data Ingestion and Integration | Airbyte | Build connections to extract, transform, and load data from multiple sources to various destinations. |
| Data Transformation | dbt | Framework to transform, test, and deploy data in warehouses. With dbt, users may define data models, transform raw data, and provide data quality checks. |
| | dbt-cloud♡ | Cloud-based platform to model, transform and analyze data in a scalable and collaborative manner. |
| Data Analysis and Visualization | Metabase | Business intelligence tool to create custom dashboards, reports, and analytics. It provides a simple and intuitive interface to ask questions and create visualizations. |
| | Superset | Enables users to make interactive dashboards. It can connect to various data sources and create visualizations to explore and analyze the data. |
| Data Orchestration | Dagster | Platform for building, deploying, and scheduling data pipelines. It integrates data from various sources and manages data transformation jobs with dependencies. |
| | Airflow | Programmatically schedule and monitor workflows in the form of Directed Acyclic Graphs (DAGs). |
| Traditional Data Processing | JupyterLab | Interactive web environment for code and visualizations. It deals with notebooks containing live code and narrative text. |
| | Excel | Spreadsheet software that allows users to create and edit data in tables, charts, and formulas. We use the open-source LibreOffice Calc instead of Microsoft Excel in our environment. |
| IT Service Management | ServiceNow♡ | Cloud-based IT service management platform that provides a suite of tools and features to streamline incident management, service catalog, asset management, and workflow automation. |
| Daily Applications | Docker, Chromium, Visual Studio Code, Bash Terminal | |

browser utility, to download the HTML files to a local directory, building all directories recursively. We also retained the directory structure of each website, as we believe the path of each document can, to some extent, represent the document's purpose. For example, the HTML files under the path "docs.getdbt.com/docs/deploy" are about deploying dbt in production or staging environments. This crawling step resulted in a total of $21,239$ HTML files.

## C.2 Filtering of HTML pages

We further filtered the crawled HTML pages based on two criteria: irrelevant content to software usage and pages containing invalid content. For the former, we mainly judged whether the page contained content related to software usage based on its path and manually confirmed it. For example, pages under "author" on the website often relate to the website developer or development team rather than software usage. Additionally, we removed category-type pages that only contained navigation information. Furthermore, we filtered out pages based on the number of tokens obtained by whitespace tokenization. We mainly removed pages with token counts less than 100, as we found that these pages predominantly contained invalid information such as access failures, invalid links, or webpage redirections. For example, the official website of `Dagster` contained numerous links to unreleased versions of documents, all of which resulted in access failures. Therefore, after removal, the number of valid pages corresponding to `Dagster` decreased from 10,065 to 332. Finally, We obtained $11,231$ filtered HTML files (see Table 8).

## C.3 HTML Preprocessing

HTML files contain a significant amount of content unrelated to the actual content of the webpage, such as "`<script>`", "`<style>`" tags, tag attributes, and developer comments. These parts may provide aesthetics to the page but are irrelevant to the document-level information. Additionally, they often occupy a large portion of the HTML file, making it excessively long for LLMs to input. To perform Retrieval-Augmented Generation (RAG) more efficiently and to help models better understand software documentation, we preprocessed these HTML files in three formats: plain text, HTML, and Markdown. These three formats of data and the original HTML files will be released to facilitate future research. The token statistics of all data formats are shown in Table 9. We describe the preprocessing details below:

**Plain Text:** We used `BeautifulSoup4` [8] to extract the textual elements from the HTML DOM [9] tree and connected these elements using "\n". This method allows us to obtain the HTML content in the simplest manner, but losing the structural information of the HTML may affect the model's understanding of the webpage content.

**Simplified HTML:** We remove all sub-trees of the HTML DOM which do not contain textual elements. We also filter out all *headers, footers, copyrights, forms, and iFrames.* We removed all HTML tag attributes since they mostly do not contain actual content or semantic information. Additionally, when a node in the HTML DOM tree has only one child node, we remove that node and directly connect its child node to its parent node. This effectively simplifies the structure and depth of the HTML. The simplified HTML preserves both the structure and content information of the original HTML with fewer tokens.

**Markdown:** We further used the `markdownify` [10] tool to convert the simplified HTML into Markdown format. Markdown format uses fewer tokens to represent structural information compared to HTML, striking a good balance between HTML and plain text formats. Moreover, since pure text includes a substantial number of newline characters used to concatenate text elements and some parts of the text content in markdown files are directly concatenated without these newlines, this results in a smaller average number of tokens in markdown files compared to the pure text format.

Concrete examples of these three formats are detailed in the task prompts (see App. I.3). In our pilot experiments (see Table 7), we compare the performances using different formats of retrieved documents on a subset (130 task samples) of Spider2-V. And pure text format outperforms the others.

---

[8]`https://beautiful-soup-4.readthedocs.io/en/latest/`

[9]The Document Object Model (DOM) is an interface that represents an HTML document as a tree structure, where each node is an object corresponding to a part of the document.

[10]`https://github.com/matthewwithanm/python-markdownify`

Table 7: Performances with different formats of retrieved documents on a subset of Spider2-V.

| RAG Format | Success Rate (%) |
|---|---|
| Pure Text | **16.92** |
| Markdown Syntax | 15.38 |
| Simplified HTML | 15.38 |

Table 8: Summary of software documentation. OrigPageNum: The number of all web pages we crawled from the documentation website. FilteredPageNum: The number of web pages obtained after filtering out irrelevant or invalid pages.

| Software | Documentation Website | OrigPageNum | FilteredPageNum |
|---|---|---|---|
| dbt/dbt-cloud | `https://docs.getdbt.com/` | 1192 | 1102 |
| Dagster | `https://release-1-7-2.dagster.dagster-docs.io/` | 10065 | 332 |
| Airflow | `https://docs.astronomer.io/` | 493 | 489 |
| Airbyte | `https://docs.airbyte.com/`
`https://airbyte.com/tutorials/`
`https://airbyte-public-api-docs.s3.us-east-2.amazonaws.com/rapidoc-api-docs.html` | 958 | 859 |
| Superset | `https://superset.apache.org/docs/` | 120 | 68 |
| Metabase | `https://www.metabase.com/docs/v0.49/`
`https://www.metabase.com/learn/` | 404 | 384 |
| Snowflake | `https://docs.snowflake.com/en/` | 4436 | 4431 |
| Bigquery | `https://cloud.google.com/bigquery/docs/` | 1330 | 1328 |
| Jupyter | `https://jupyterlab.readthedocs.io/en/4.1.x/` | 2241 | 2238 |
| **Total** | | **21239** | **11231** |

# D  Details of Executable Environment in Spider2-V

In this section, we briefly introduce OSWORLD [37] and how we adapt it to meet our requirements.

## D.1  Overview

Spider2-V formalizes the interaction with a Ubuntu desktop as a partially observable Markov decision process (POMDP) $(\mathcal{S}, \mathcal{O}, \mathcal{A}, \mathcal{T}, \mathcal{R})$ with state space $\mathcal{S}$, observation space $\mathcal{O}$, action space $\mathcal{A}$, state transition function $\mathcal{T} : \mathcal{S} \times \mathcal{A} \rightarrow \mathcal{S}$ and reward function $\mathcal{R} : \mathcal{S} \times \mathcal{A} \rightarrow \mathbb{R}$. Given the current observation $o_t \in \mathcal{O}$ from the desktop, the agent needs to predict action $a_{t+1} \in \mathcal{A}$ for the next step. An admissible action incurs a change in the latent state space $s_{t+1} \in \mathcal{S}$, and the environment feedback $o_{t+1}$. The interaction loop repeats until a special "DONE" or "FAIL" action is issued, wherein the task episode ends and a reward $r = \mathcal{R}(s_T) \in \{0, 1\}$ is computed, with 1 indicating task success.

The executable computer environment (a Ubuntu operating system) is built upon virtual machines (VMs). By using the "*snapshot*" functionality of VM, the localhost environment state can be

Table 9: Average number of page tokens of different documentation formats. We used `TikToken`, a fast BPE tokenizer for use with OpenAI's models, to calculate the token count for gpt-3.5-turbo.

| Software | OrigHTML | PlainText | SimpHTML | Markdown |
|---|---|---|---|---|
| dbt/dbt-cloud | 17954 | 1669 | 2963 | 1510 |
| Dagster | 131777 | 2615 | 4704 | 2290 |
| Airflow | 35011 | 2007 | 3885 | 1829 |
| Airbyte | 30124 | 2448 | 4328 | 2329 |
| Superset | 10798 | 1398 | 2389 | 1415 |
| Metabase | 33523 | 2288 | 4690 | 2333 |
| Snowflake | 105155 | 1750 | 3342 | 1595 |
| Bigquery | 103748 | 6245 | 11777 | 5718 |
| Jupyter | 224153 | 11240 | 19917 | 6743 |
| **Total** | **109119** | **4273** | **7789** | **3212** |

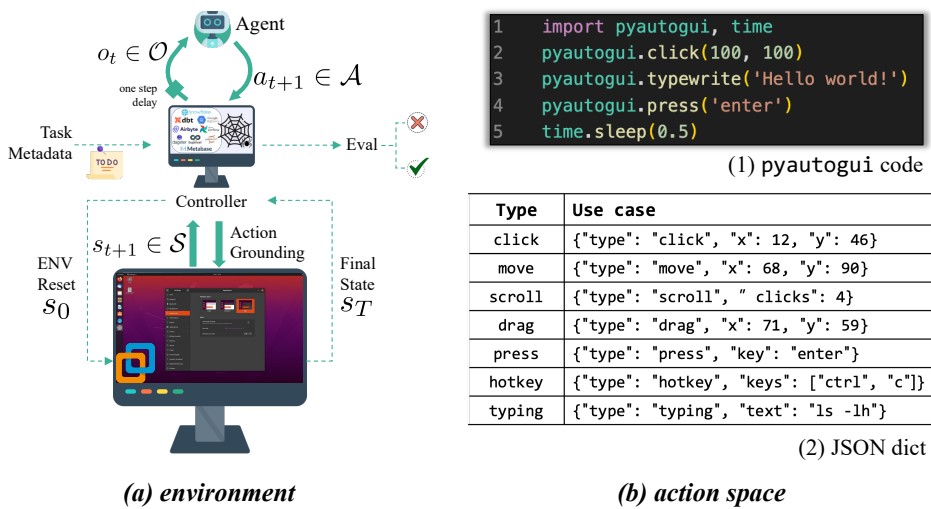

*(a) environment*

```
1   import pyautogui, time
2   pyautogui.click(100, 100)
3   pyautogui.typewrite('Hello world!')
4   pyautogui.press('enter')
5   time.sleep(0.5)
```

(1) `pyautogui` code

| Type | Use case |
|---|---|
| click | {"type": "click", "x": 12, "y": 46} |
| move | {"type": "move", "x": 68, "y": 90} |
| scroll | {"type": "scroll", " clicks": 4} |
| drag | {"type": "drag", "x": 71, "y": 59} |
| press | {"type": "press", "key": "enter"} |
| hotkey | {"type": "hotkey", "keys": ["ctrl", "c"]} |
| typing | {"type": "typing", "text": "ls -lh"} |

(2) JSON dict

*(b) action space*

Figure 9: Overview of the executable environment of Spider2-V and two types of action space.

completely recovered to a stored history state. This snapshot with task-specific setup functions (see § 2.2) serve as the initial state $s_0 \in \mathcal{S}$ for different tasks. And a core *controller* is responsible for grounding action $a_t$ (see App. D.2) into the VM desktop and obtaining observations $o_t$ (see App. D.3) from the resulting state of VM. After the agent issues a special action "`DONE`" or "`FAIL`", the controller will invoke the customized evaluation function for the current task (see § 2.3) and report the metric score. The entire procedure is shown in Figure 9(a).

## D.2 Action Space

For generic actions that support both CLI and GUI, we introduce two different action spaces:

**pyautogui code**  This action space accepts arbitrary executable python code. Particularly, code snippets that using python library "`pyautogui`" to control the mouse and keyboard are strongly recommended. Generally, mouse-based actions (*e.g.*, click and scroll) directly manipulate the GUI screen, while keyboard-based actions (*e.g.*, typewrite and hotkey) interact with the CLI such as the bash terminal and code editor (*e.g.*, Visual Studio Code).

**JSON dict** Inspired by the "pyautogui" library, we summarize 7 actions to simplify the action space. This small set can cover all CLI and GUI actions needed on the desktop. For each action and its parameters, we further encapsulate it into a JSON dict to restrict the output format. The API specification and use cases are formally described in prompt messages (see App. I.1.2). And the checklist of all 7 actions is presented in Figure 9(b).

## D.3 Observation Space

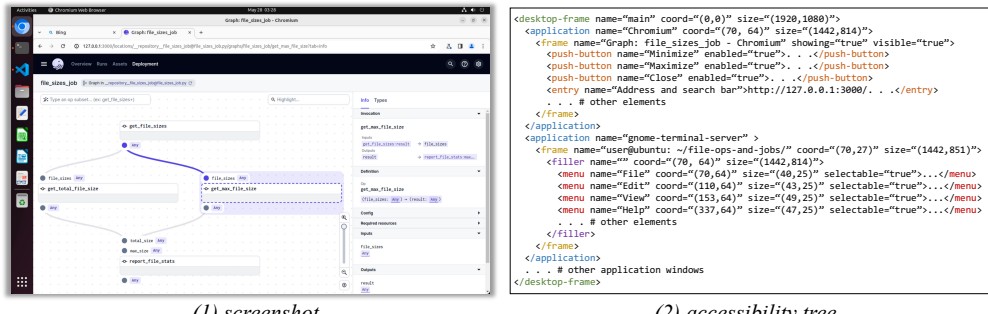

*(1) screenshot*      *(2) accessibility tree*

Figure 10: Two observation types: screenshot and accessibility tree (`a11ytree`).

With respect to observations, there are two widely used alternatives (see Figure 10): 1) image-style screenshot of the entire desktop, and 2) text-format accessibility tree (`a11ytree`). The accessibility tree, obtained from the Assistive Technology Service Provider Interface (ATSPI) library [11], is a text-format abstraction of the entire computer desktop which describes the name, type, status (*e.g.*, a menu bar is "*selectable*"), position (*e.g.*, in Figure 10 (2), the attributes "*coord*" and "*size*" together define the rectangle position), and text content embedded in each element (e.g., windows, panels, buttons, and input boxes). We extract `a11ytree` using python library `pyatspi` and convert it into the XML format. It functions similar to DOM (Document Object Model) tree for websites.

### D.3.1 Two tricks: Set-of-Mark and Execution Feedback

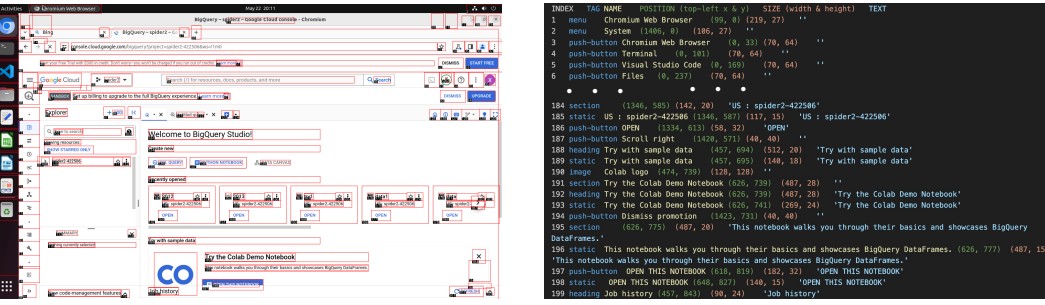

Figure 11: Screenshot with bounding boxes.      Figure 12: Converted table of `a11ytree`.

Figure 13: Illustration of the aligned observation type set-of-mark (SoM).

**Set-of-Mark (SoM)** The original text-style accessibility tree (`a11ytree`) and image-style screenshot do not align with each other. To compensate for this deficiency, we follow OSWORLD [37] and WebArena [48] to draw bounding boxes for elements of interest in the screenshot and label these elements with numeric indexes. The accurate coordinates of these bounding boxes are extracted from the `a11ytree`. Furthermore, we re-organize the `a11ytree` into a table (each leaf node in `a11ytree` is converted into one row) and insert another attribute/column "`index`" for each node in the tree. The value of attribute "`index`" is exactly the numeric label of the corresponding element in the screenshot. The aligned screenshot and `a11ytree` (*a.k.a.*, set-of-mark, SoM [39]) are illustrated in Figure 13.

---

[11] https://docs.gtk.org/atspi2/

**Examples of Execution Feedback Messages**

```
Here are failed actions with their error messages in your last response:
# Action 1
import pyautogui
index_34 = (23, 43)
pyautogui.click(index_343)
# Execution error:
Traceback (most recent call last):
NameError: name 'index_343' is not defined

# Action 2
import pyautogui
import time
pyautogui.write('USE DATABASE IMDB\n\\n')
# Execution error:
File "<string>" line 3
pyautogui.write('USE DATABASE IMDB
                    ^
SyntaxError: unterminated string literal
```

**Execution Feedback**   We also incorporate another type of information as the observation, namely the *execution feedback* of actions (see messages above). We notice that, some predicted actions may be parsed erroneously or fail to be executed. In this case, the two observation types mentioned before are not changed at all. And the agent repeatedly urges to conduct the same incorrect action. To inform the agent of execution errors, we include this execution feedback as the third observation type.

# E   Details of Experimental Resources

For those closed-source models, we directly resort to their API services with the following versions `gpt-4o-2024-05-13`, `gpt-4-1106-vision-preview`, `gemini-1.5-pro-001` and `claude-3-opus-20240229`, respectively. Note that, results could be updated from time since they are close-sourced. While for open-weight models, we use the GroqCloudTM API platform [10] where the hardware resources are provided by Groq.

With respect to the runtime complexity and API cost per task, they are tightly connected to the maximum interaction turns, trajectory window size and network delay when calling APIs. Using the detailed configuration below (also the most powerful agent in main experiments),

- action space = pyautogui code,
- observation space = Set-of-Mark (prompt tokens include image pixels),
- maximum interaction turn = 15, trajectory window size = 3,
- w/ retrieval augmentation, top_k = 4, chunk_size = 512 tokens,

the average cost with GPT-4o is shown in Table 10 below:

Table 10: Average running time and cost per task with GPT-4o.

| Model | *Avg.* **Running Time** | *Avg.* **Prompt Tokens** | *Avg.* **Completion Tokens** | *Avg.* **Cost Per Task** |
|-------|------------------------|--------------------------|------------------------------|--------------------------|
| GPT-4o | 8.7 minutes | 0.32M | 1.6K | $1.64 |

As for the environment delay, the average running time for each task can be further decoupled into:

1. environment setup time, that is calling a series of functions defined in § 2.2,

2. response time when calling APIs or querying LLMs (may depend on the network traffic),

3. time delay to perform each action in the computer desktop (this is controllable, and we set the time delay to 1 second for the action to take effect),

4. time delay to obtain the screenshot and accessibility tree (observations) after each action is grounded in the virtual machine,

5. evaluation time at the end of the episode when we invoke task-specific scripts introduced in § 2.3.

According to Table 11, we can find that the majority time is spent on multi-turn API calls (calculated by elimination) due to long prompt tokens and slow inference time.

Table 11: Decomposition of *avg.* running time per task with GPT-4o API.

| Environment Setup | Delay Per Action | Delay Per Observation | Evaluation Time |
|:---:|:---:|:---:|:---:|
| $\sim 1$ minute | 1 second | $2 \sim 8$ seconds | $< 1$ minute |

## F  Limitations and Social Impact

**Limitations**   Despite the contribution of Spider2-V, there are still several limitations: a) One concern is about the scalability of annotating tasks. The complicated annotation procedure (elaborated in § 2.2, § 2.3 and § 3.1) requires professional expertise and tool-specific programming skills. Annotators need to understand how to manipulate the professional applications, manually design the task, and write instructions, environment setup methods and evaluation functions. This may prevent large-scale crowd-sourcing among non-experts. b) Another major issue is the interaction efficiency. To resolve real-world tasks in Spider2-V, LLM agents need multiple turns of interaction with the executable environment. During the interaction process, the prompt length and API cost increase dramatically if we want to insert more history context. However, we hope this arising problem can inspire more interesting work in improving the agent interaction efficiency. c) Lastly, the evaluation on tasks involving authentic user accounts in Spider2-V may encounter failures due to unexpected errors caused by the remote cloud-hosted enterprise, e.g., the web server suddenly crashes.

**Social Impact**   This paper proposes Spider2-V, the first-of-its-kind multimodal agent benchmark focusing on professional data science and engineering workflows, featuring 494 real-world tasks in authentic computer environments and 20 enterprise-level professional applications. The goal is to enhance productivity for data scientists and lay the groundwork for developing practical multimodal agents that can revolutionize the automation of data science and engineering workflows. The potential negative impact is that, once more effective agents are delivered for this type of task set, it could lead to some degree of unemployment among data scientists and engineers. Besides, for tasks involving authentic user accounts, benchmark users need to create authentic accounts to exploit the cloud-hosted sandbox services and test their agents. It potentially increases the management burden for software service providers. Furthermore, since the interaction proceeds in a fully realistic environment, the developed universal digital agent could be used to bypass CAPTCHA systems in the future.

## G  Format of Task Examples

In this section, we briefly introduce the data format of tasks. Each task instance is represented as a JSON dictionary which contains the following fields: (see Listing 1 for a realistic example)

- `id`: globally unique uuid of the current task example.
- `instruction`: the task instruction which indicates the task goal.
- `source`: a list of referenced tutorial links to construct the current task.
- `config`: a list of dictionaries which defines the sequential operations to initialize and reset the computer desktop. Each dictionary contains the function name (the "`type`" key) and its parameters (the "`parameters`" key), indicating one environment setup function. For example, in Listing 1, we define 3 environment reset functions, namely
  - `bigquery_init`: clear the cloud workspace of Google project "`bigquery-project`";

- ■ `google_chrome_browser`: launch the Google Chrome application in the virtual machine;
- ■ `bigquery_login`: simulate the Google account login operation with playwright.
- • `related_apps`: a list of application names which should be used in the current task.
- • `tags`: a list of tags denoting different categories for this task.
- • `evaluator`: a dictionary containing 3 fields: `func`, `result`, `expected`. It defines how to evaluate the final results once task completion. Concretely,
  - ■ the "`func`" field defines the name of our customized function (or metric) which is used to compare the predicted result and the expected golden result;
  - ■ the "`result`" field defines how to extract the predicted result from the final open-ended environment states, *e.g.*, copied from a specific file path in the virtual machine;
  - ■ the "`expected`" field defines how to obtain the golden result, *e.g.*, from a backup local file.

For example, in Listing 1, we utilize the function "`compare_csv`" to compare the predicted file "`/home/user/Downloads/answer.csv`" in the virtual machine and the golden file "`answer_gold.csv`" in local host.

Listing 1: BigQuery Task Example of JSON Format from Spider2-V

```
1  {
2    "id": "3363a913-d3e9-42c2-9d76-9cd9e9bafec7",
3    "instruction": "I want to know how many austin bike stations are
          ↪ active? Save the SQL execution results into 'answer.csv'
          ↪ under '/home/user/Downloads/' folder.",
4    "source": [
5      "https://cloud.google.com/bigquery/docs/quickstarts/query-public-
          ↪ dataset-console"
6    ],
7    "config": [
8      {
9        "type": "bigquery_init",
10       "parameters": {
11         "config_file":
              ↪ "evaluation_examples/settings/google/gcp_config.json",
12         "project_name": "bigquery-project",
13         "actions": [{"type": "empty"}]
14       }
15     },
16     {
17       "type": "google_chrome_browser",
18       "parameters": {
19         "debugging_port": 1337,
20         "listening_port": 9222
21       }
22     },
23     {
24       "type": "bigquery_login",
25       "parameters": {
26         "settings_file":
              ↪ "evaluation_examples/settings/google/settings.json",
27         "config_file":
              ↪ "evaluation_examples/settings/google/gcp_config.json",
28         "project_name": "bigquery-project"
29       }
30     }
31   ],
32   "related_apps": ["bigquery", "chromium"],
33   "tags": ["cli+gui", "account", "data_warehousing"],
34   "evaluator": {
35     "func": "compare_csv",
36     "result": {
37       "type": "vm_file",
38       "path": "/home/user/Downloads/answer.csv",
```

```
39          "dest": "answer.csv"
40        },
41        "expected": {
42          "type": "local_file",
43          "path":
              ↪  "evaluation_examples/examples/bigquery/3363a913-d3e9-42c2
              ↪  -9d76-9cd9e9bafec7/answer_gold.csv",
44          "dest": "answer_gold.csv"
45        }
46      }
47   }
```

**Tasks with Authentic User Accounts**    One hurdle of utilizing professional enterprise software is authentic user accounts. Accordingly, the entire task set is split into two parts:

1)  *w/o authentic user accounts*: In Spider2-V, more than half of the tasks (66%) can be launched in localhost. That is, we can start the Web server through Docker containers and leverage the default login account. Benchmark users do not need to register a new one and the task can be completed totally in localhost. They can choose this dataset split merely for easier reproduction.

2)  *w/ authentic user accounts*: For the other 34% tasks, we provide a ".json" template file for each professional application which requires a real-world user account (*e.g.*, BigQuery, ServiceNow, etc.). If benchmark users want to evaluate on them, to ensure successful automatic setup and evaluation, they need to register each account manually and fill personal credentials into different .json templates (in total, 6 templates/software across Spider2-V). Once the credentials are filled, the automatic environment setup and customized evaluation for each task will perform the remaining work. This works similar to setting the OpenAI [23] API key. And previous work WorkArena [6] also functions in this way. For each software, we also provide a detailed step-by-step tutorial on how to obtain each json field in the Github repository. A simple glance over the Snowflake template is provided in Listing 2.

Listing 2: JSON Template for Snowflake Account

```
1   {
2       "account": "https://{xxxxxxxx}.snowflakecomputing.com",
3       "user": "USER_NAME",
4       "password": "YOUR_PASSWORD"
5   }
```

With respect to credit card payment, all tasks do not need additional payments throughout the entire process. Because Spider2-V only exploit the Sandbox [9] or Trial [31] accounts, which means the free tier can adequately cover all consumption over the entire benchmark. Users even do not need to add the credit card when signing up.

# H Task Examples

In this part, we present diverse examples in Spider2-V.

Table 12: Real task examples from Spider2-V.

| Related App(s) | Instruction | Screenshot After Initialization |
|---|---|---|
| Dagster dbt Chromium VS Code | *I have a dbt project "jaffle_shop". Please integrate this project into dagster and add a dagster asset "customers" according to the schema provided by the file "~/dbt-dagster-project/jaffle_shop/customers_schema.yml". Materialize the asset in the opened dagster UI.* |  |
| BigQuery Chromium | *I have just uploaded data about Ameraican babies into table 'names_2014'. I am curious about the top five names for US babies that were assigned male at birth in that year. Please save the 'name' and 'count' into another table 'top5_male_2014' in the same dataset for me.* |  |
| Dagster Airflow MySQL Chromium VS Code Terminal | *I have defined an Airflow DAG. Please help me migrate it to Dagster based on the requirements in "README.md". Remember to launch the Dagster webserver from "dagster_migration.py" and start the DAG schedule. Test the schedule on Dagster UI Launchpad and make sure the job can succeed.* |  |
| Metabase Chromium | *I want to have a stack bar chart out of Sample Database in metabase. Could you help me visualize the data of Products table and summarize the data of Sum of price by Product Category and Created At - Quarter. Then stack the visualized chart. Please help me download the visualization as a PNG file, and rename it to "stack_chart.png".* |  |
| Jupyter Chromium | *I want to use Logistic Regression to predict whether a student will be admitted to a college or not, and have now built the code framework in this open jupyter notebook. Please read the framework code and complete all the #TODO sections. Finally, you need to run the code and save it.* |  |

*Continued on next page*

| Related App (s) | Instruction | Screenshot After Initialization |
|---|---|---|
| Excel | *Add a new column named "Profit" and calculate the profit for each week by subtracting "COGS" from "Sales" in that column.* |  |
| Superset Chromium | *Help me create a rolling mean line chart for table flights to see the trend of the average cost per day. The rolling period should be 7 and save the chart as the name "rolling_mean".* |  |
| Airbyte Chromium | *Help me set up the destination of data transfer to a local JSON file in the Airbyte local UI. The target file path is /local/json_destination.* |  |
| dbt-cloud Chromium | *I've created an empty dbt cloud project named "test_connection". Could you help me set up the connection to a BigQuery GCP? You don't need to configure the repository for the project, and the credential file is provided at desktop.* |  |
| Airflow Docker VS Code Chromium | *I have defined two DAGs to fetch and process data from TheCocktailDB. I hope to change the schedule of the consumer DAG such that each time the resulting files of the producer are updated, the consumer DAG is triggered. Can you help me with this data-aware scheduling?* |  |
| Dagster Chromium VS Code | *Modify the current Dagster machine learning pipeline by adding two features "Age" and "Fare" to the Logistic Regression model from the data (you should fill in the NaN values by the mean of the column). Launch a run of the job "sklearn_job", and schedule it to run at every hour on weekdays.* |  |

*Continued on next page*

| Related App (s) | Instruction | Screenshot After Initialization |
|---|---|---|
| Snowflake Chromium | *I heard there are many free to download datasets on Snowflake marketplace. And I am really curious about worldwide addresses. Could you help me get one database about it? Name it 'WORLD-WIDE_ADDRESSES'.* |  |
| ServiceNow Chromium | *Go to the hardware store and order 8 "iPad mini" with configuration {'Choose the colour': 'Purple', 'Choose the storage': '256'}* |  |
| BigQuery Chromium | *Load the data from the Google drive Spider002 folder into Bigquery's 'data1' table of 'information' datasets.* |  |
| Metabase Postgresql Chromium | *Help me finish the metabase login setup with information shown in setup.json.* |  |
| Dagster Chromium VS Code | *I just built a 3-step Dagster pipeline. Now, I want to run it regularly to keep all assets up to date. Name the target job 'hacker_news_pipeline' and schedule it to run every hour.* |  |
| dbt-cloud Chromium Terminal | *Install dbt-cloud-cli from GitHub and extract the binary to the same folder as the dbt project "analytics". Follow the instruction "Step 1: Install" specified in the opened account profile page.* |  |

# I Prompts for Multi-modal Agents

Multi-modal agent baseline involves complex prompt engineering. The following sections will introduce the system prompt, task prompt, and retrieved context augmented prompt.

## I.1 System Prompt

The entire system prompt consists of the environment prompt, observation space prompt, action space prompt, and general tips. Different action/observation types have different prompts. In this section, we will introduce each one in turn and present the overall system prompt at last.

### I.1.1 Observation Space Prompt

The four different observation space settings, namely 1) screenshot, 2) a11ytree, 3) screenshot+a11ytree, and 4) SoM, each has a different prompt.

**Screenshot Setting**

```
After each action step, you will get an image-style observation,
↪   which is the screenshot of the computer screen. And you need to
↪   predict the next action on the computer based on this image.
```

**Accessibility Tree Setting**

```
After each action step, you will get a text-style observation, which
↪   is extracted and pruned from the accessibility tree based on
↪   AT-SPI library. The accessibility tree describes the elements
↪   (e.g., panels, icons, buttons, frames, windows, applications) on
↪   the computer desktop, as well as its embedded text content,
↪   status and positions. For simplicity, we prune the original tree
↪   and only extract useful information into a tabular format for you.
↪   Here is a quick glance on the observation:
TAG, NAME, POSITION (top-left x & y), SIZE (width & height), TEXT
menu, Visual Studio Code, (99, 0), (184, 27), ''
push-button, Chromium Web Browser, (0, 33), (70, 64), ''
terminal, Terminal, (70, 74), (1430, 832), '(base)
↪   user@ubuntu:~/projects/$'

... more rows ...

, where `TAG` / `NAME` is the element type / name respectively.
↪   `POSITION` and `SIZE` together describe the square position of
↪   this element on the computer screen. For example, if you want to
↪   click one button, you can click any point in the square area
↪   defined by `POSITION` and `SIZE`. Assume that the position of
↪   this button is (100, 200), and the size is (40, 40), the CENTER
↪   of this button is (120, 220), which is calculated by x = 100 + 40
↪   / 2 = 120, y = 200 + 40 / 2 = 220. `TEXT` refers to the text
↪   content embedded in the element, e.g., the bash terminal output
↪   or texts in an editable input box.

And you will predict the next action of the computer based on the
↪   accessibility tree.
```

**Screenshot + Accessibility Tree Setting**

```
The observation space is a combination of two sources: 1) image-style
↪   screenshot of the desktop, and 2) text-style accessibility tree
↪   derived from AT-SPI library.

### Screenshot

After each action step, you will get a image-style observation, which
↪   is the screenshot of the computer screen. And you need to predict
↪   the next action on the computer based on this image. You can use
↪   this image to locate the elements on the screen or check the
↪   status of the computer, especially whether the previous action is
↪   successful or not.

### Accessibility Tree

The accessibility tree describes the elements (e.g., panels, icons,
↪   buttons, frames, windows, applications) on the computer desktop,
↪   as well as its embedded text content, status and positions. For
↪   simplicity, we prune the original tree and only extract useful
↪   information into a tabular format for you. Here is a quick glance
↪   on the observation:

TAG, NAME, POSITION (top-left x & y), SIZE (width & height), TEXT
menu, Visual Studio Code, (99, 0), (184, 27), ''
push-button, Chromium Web Browser, (0, 33), (70, 64), ''
terminal, Terminal, (70, 74), (1430, 832), '(base)
↪   user@ubuntu:~/projects/$'

... more rows ...

, where `TAG` / `NAME` is the element type / name respectively.
↪   `POSITION` and `SIZE` together describe the square position of
↪   this element on the computer screen. For example, if you want to
↪   click one button, you can click any point in the square area
↪   defined by `POSITION` and `SIZE`. Assume that the position of
↪   this button is (100, 200), and the size is (40, 40), the CENTER
↪   of this button is (120, 220), which is calculated by x = 100 + 40
↪   / 2 = 120, y = 200 + 40 / 2 = 220. `TEXT` refers to the text
↪   content embedded in the element, e.g., the bash terminal output
↪   or texts in an editable input box.

You can use the accessibility tree to accurately locate positions of
↪   useful elements on the screen and check the concrete textual
↪   contents of elements.

By combining the screenshot and accessibility tree, you should be
↪   intelligent to predict the next feasible and meaningful action.
```

**SoM Setting**

The observation space is a combination of two sources: 1) image-style
↪    screenshot of the desktop with interact-able elements marked with
↪    numerical indexes, and 2) text-style accessibility tree derived
↪    from AT-SPI library.

### Labeled Screenshot

After each action step, you will get a image-style observation, which
↪    is the screenshot of the computer screen. For ease of locating
↪    positions of elements, we extend the original screenshot with
↪    index marks. That is, some salient elements which can be
↪    interacted with (e.g., a button or editable input box) are marked
↪    with line boudaries and numeric indexes. You can use this image
↪    to locate the elements on the screen or check the status of the
↪    computer, especially whether the previous action is successful or
↪    not.

### Accessibility Tree

The accessibility tree describes the elements (e.g., panels, icons,
↪    buttons, frames, windows, applications) on the computer desktop,
↪    as well as its embedded text content, status and positions. For
↪    simplicity, we prune the original tree and only extract useful
↪    information into a tabular format for you. Here is a quick glance
↪    on the observation:
INDEX, TAG, NAME, POSITION(top-left x & y), SIZE(width & height),TEXT
1,    menu,        Visual Studio Code, (99, 0),   (184, 27),   ''
2,    push-button, Chromium Web Browser, (0, 33),     (70, 64),   ''
3,    terminal,    Terminal,            (70, 74),   (1430, 832), (base)
user@ubuntu:~/projects/$'
... more rows ...

, where `INDEX` indicates exactly the numeric label for each element
↪    marked in the screenshot. You can use this alignment information
↪    to simplify your predicted action. For example, you can use
↪    `pyautogui.click(index_2)` to represent clicking the CENTER of
↪    the element with index 2 on the screenshot. We will automatically
↪    perform the position calculation and substitution for you. `TAG`
↪    / `NAME` is the element type / name respectively. `POSITION` and
↪    `SIZE` together describe the square position of this element on
↪    the computer screen. For example, if you want to click one button,
↪    you can click any point in the square area defined by `POSITION`
↪    and `SIZE`. Assume that the position of this button is (100, 200),
↪    and the size is (40, 40), the CENTER of this button is (120, 220),
↪    which is calculated by x = 100 + 40 / 2 = 120, y = 200 + 40 / 2 =
↪    220. `TEXT` refers to the text content embedded in the element,
↪    e.g., the bash terminal output or texts in an editable input box.
You can use the accessibility tree to accurately locate positions of
↪    useful elements on the screen and check the concrete textual
↪    contents of elements.
By combining the screenshot and accessibility tree, you should be
↪    intelligent to predict the next feasible and meaningful action.

### I.1.2 Action Space Prompt

As for the prompt of action space, we provide two choices: 1) pyautogui code, and 2) JSON dict.

**pyautogui Code**

```
You are required to use `pyautogui` to perform the action grounded to
↪  the observation. And the action space includes two types:

1. Python code block using pyautogui wrapped by 3 backticks, e.g.,
```python
# you python code here, e.g.,
pyautogui.hotkey('ctrl', 'c')
```

2. Three pre-defined special actions: [WAIT, FAIL, DONE]
- When you think you have to wait for some time, return ```WAIT```;
- When you think the task can not be done, return ```FAIL```, don't
↪  easily say ```FAIL```, try your best to do the task;
- When you think the task is done, return ```DONE```.
These 3 actions also need to be wrapped by 3 backticks.

### REMEMBER THAT:

0. We will import libraries `pyautogui` and `time` automatically for
↪  you, but if you use other python libraries, PLEASE IMPORT THEM
↪  FIRST ALTHOUGH THIS IS DISCOURAGED;
1. DONOT use the `pyautogui.locateCenterOnScreen` function to locate
↪  the element you want to operate with, since we have no image of
↪  the element you want to operate with;
2. DONOT use the `pyautogui.screenshot` function to make screenshot;
3. For time efficiency, you can return one line or multiple lines of
↪  python code to perform continuous actions in one response. For
↪  example, your response may contain the following code block:
```
pyautogui.moveTo(100, 210)
pyautogui.dragTo(500, 200, button='left', mouseDownUp=True)
pyautogui.rightClick()
```
4. When predicting multiple lines of code, make some small delay like
↪  `time.sleep(0.5)` interval, such that the machine can response
↪  correctly. And it is STRONGLY RECOMMENDED that, for one action
↪  which may influence the environment significantly (e.g., click
↪  the button of one application to open it, or click a web link
↪  which navigates to a new page), it is better to predict this
↪  action without follow-ups in order to observe the changes in
↪  environment states first;
5. Each time when you predict code, neither variables nor function is
↪  shared acrossed different code blocks. In other words, each code
↪  block will be executed in isolation;
6. For coordinates (x, y), please speculate or calculate by yourself
↪  based on the observation of previous interaction turn. BE CAREFUL
↪  to ensure the coordinates are feasible.
7. Please pay attention that, code wrapped by 3 backticks ``` will be
↪  recognized as an action in the action space. Therefore, when you
↪  output non-action code, please use other symbols like '''
↪  instead.
```

**JSON Dict (truncated)**

```
Firstly, we use json dict to describe the types and parameters for
↪   each action we allowed (`required=true` means this argument must
↪   be provided). Then, we demonstrate use cases, and precautions.

### Specification for All Actions

ACTION_LIST = [
    {
        "action_type": "MOVE_TO",
        "note": "move the cursor to a specified position (x, y)",
        "parameters": {
            "x": {
                "type": float,
                "range": [0, MAX_SCREEN_WIDTH],
                "required": true,
            },
            "y": {
                "type": float,
                "range": [0, MAX_SCREEN_HEIGHT],
                "required": true,
            }
        }
    },
    ... more action dicts ...
]

### Use Cases

- For MOVE_TO, you need to predict the x and y coordinate of the
↪   mouse cursor, the left top corner of the screen is (0, 0).
Use case: move the mouse to position (56.1, 65.0)
```json
{
    "action_type": "MOVE_TO",
    "x": 56.1,
    "y": 65.0
}

... more use cases ...

### Precautions

1) The output action MUST BE CHOSEN and CAN ONLY BE CHOSEN from the
↪   action space (json dict) defined above, otherwise your action
↪   will be considered as invalid and you will get a penalty. For
↪   example, bash, sql, or python code WILL NOT be executed;
2) For each action dict, STRICTLY OBEY THE FORMAT, which must contain
↪   the `action_type` field and required parameters. Optional
↪   parameters will be set to default values if not provided. NEVER
↪   RETURN ME ANYTHING ELSE WHICH IS NOT DEFINED;
3) For efficiency, you CAN predict multiple actions in one response,
↪   but REMEMBER TO WRAP EACH ACTION DICT SEPARATELY using backticks
↪   ```json and ```.
```

### I.1.3 Overall System Prompt

```
You are an intellignet agent who is expert in completing data
science/engineering tasks using professional tools on computer. You
↪ have deep understanding of computer basics and data
↪ science/engineering knowledge.
Now, you will interact with a real desktop environment, which is an
↪ Ubuntu operating system that has access to the Internet. You
↪ should strictly follow the user instruction, communicate with the
↪ environment and try your best to complete the given data-related
↪ task successfully. Generally, you will communicate with the
↪ environment in this interactive and continuous manner:
1) In each iteration, you should take one action to control the
↪ keyboard or mouse in the desktop environment given the actions
↪ and observations from a few previous steps;
2) Then, you will obtain new observations from the environment after
↪ the action is grounded (you do not need to worry about the
↪ execution, we will perform it for you);
3) Repeat steps 1) and 2) until you think the work is done.

Here are the details of the action spaces (including usage and
↪ precautions) and observation spaces:

{{action_prompt}}

{{observation_prompt}}

Besides, here are some important tips for you to better complete the
↪ task:
1. My computer's password is 'password', feel free to use it when you
↪ need sudo rights.
2. The screen size for the running desktop is: ({screen_width},
↪ {screen_height}).
3. Some action may need time to reflect in the environment (e.g.,
↪ code execution and web page loading), please be patient and refer
↪ to the WAIT action.
4. Try to complete the task in as few steps as possible, we are on a
↪ tight budget.
5. Try to use the applications we opened for you as possible, e.g.,
↪ use the opened gnome-terminal instead of the embedded one in
↪ Visual Studio Code.
6. For critical actions (e.g., opening an application or clicking a
↪ button), ensure the action succeeds before predicting or
↪ proceeding to the next one. That is, DO NOT be greedy to predict
↪ all actions all at once in one response without confirming the
↪ observation of a significant action.
7. When you try to write codes or texts, please ensure you have
↪ focused on the right window or input panel. If the input panel
↪ already has some texts, be careful that you may need to clear or
↪ selecting them before overwritting.
8. DO NOT be stubborn to complete the task in one step. You can break
↪ down the task into several steps and complete them one by one.
9. DO NOT be stupid to repeat the same actions without any progress.
↪ If you find that the action is not effective in the observation,
↪ try another one.
10. RETURN ME ONLY THE ACTION DEFINED IN ACTION SPACES. NEVER EVER
↪ RETURN ME ANYTHING ELSE. THIS IS CRITICAL!!!
```

## I.2   Task Prompt

The task instruction for Spider2-V has two forms. The abstract instruction describes the overall goal of a task without a step-by-step solution, thus testing both planning and grounding abilities. The verbose instruction provides a detailed tutorial-like solution to the task, primarily validating the grounding ability.

### I.2.1   Example of Task Prompt for Abstract Instructions

```
Now, let's start the task!
You are asked to complete the following task: I want to build an
↪  airflow project connecting to a local postgres database. Could
↪  you install docker, astro and postgresql for me. The sudo
↪  password is 'password' (' not included). By the way, configure
↪  docker and postgresql to auto-start on boot, and allow me to
↪  prevent typing sudo when using docker each time.
```

### I.2.2   Example of Task Prompt for Verbose Instructions

```
Here is a step-by-step tutorial from an expert instructing you how to
↪  complete it:

Now we want to upload data from xlang_gcs/google_ads/ in google cloud
↪  storage to my dataset google_ads. To do this:
1. Click the "+ ADD" button next to the "Explorer" panel.
2. Click the "Google Cloud Storage" panel on the pop-up window.
3. In the input box "Google Cloud Storage", enter the
↪  'xlang_gcs/google_ads/account_history_data.csv' in the second
↪  windows. This window is labeled 'Select file from GCS bucket or
↪  use a a URI pattern'.
4. Destination Part, set Dataset to 'my_google_ads'
5. In Destination Part, set Table to 'account_history_data'
6. In Schema part, Mark the check mark in front of Auto detect.
7. Then, click the blue `CREATE TABLE` button at the bottom.
8. After page loading, click the "+ ADD" button next to the
↪  "Explorer" panel again.
9. Click the "Google Cloud Storage" panel on the pop-up window.
10. In the input box "Google Cloud Storage", enter the
↪  'xlang_gcs/google_ads/account_stats_data.csv' in the second
↪  windows. This window is labeled 'Select file from GCS bucket or
↪  use a a URI pattern'.
11. Destination Part, set Dataset to 'my_google_ads'
12. In Destination Part, set Table to 'account_stats_data'
13. In Schema part, Mark the check mark in front of Auto detect.
14. Click the `CREATE TABLE` button at the bottom left in the pop-up
↪  window.
Eventually, we have completed this task.

You can exactly follow the detailed plan above or proactively tackle
↪  the task based on the real-time environment interaction by
↪  yourself.
```

### I.3 Example of Retrieved Context Augmented Task Prompt

We also introduce a RAG setting, where we collect and clean the official documents of the professional tools as the retrieval corpus. We select top $k$ ($k$ may depend on the constraint on input length) chunks (each chunk is a token sequence with maximum length $512$) and insert them into the prompt input. Here are three demonstrations of different formats of the retrieved context.

**Pure Text Format**

```
We also retrieve relevant documentation from the web to help you with
↪   the task:

Documentation Source:
release-1-7-2.dagster.dagster-docs.io/integrations/dagstermill/using-
↪   notebooks-with-dagster.html

Documentation Title:
Using Jupyter notebooks with Papermill and Dagster Tutorial

Documentation Content:
The page will display the notebook asset in the Asset Graph.
If you click the notebook asset, a sidebar containing info about the
↪   asset will slide out from the right side of the page. In the
↪   Description
section of the panel is a View Source Notebook button:
This button allows you to view the notebook directly in the UI. When
↪   clicked, Dagster will render the notebook - referenced in the
notebook_path parameter - that'll be executed when the
↪   iris_kmeans_jupyter asset is materialized:
Click the Materialize button. To view the execution as it happens,
↪   click the View button in the alert that displays.
After the run completes successfully, you can view the executed
↪   notebook in the UI. Click the asset again and locate the View
↪   Notebook button in the Materialization in Last Run section of the
↪   sidebar:
Click the button to display the executed notebook - specifically, the
↪   notebook that was executed and written to a persistent location:
Step 5: Add an upstream asset #
While our iris-kmeans notebook asset now materializes successfully,
↪   there are still some improvements we can make. The beginning of
↪   the notebook fetches the Iris dataset, which means that every
↪   time the notebook is materialized, the data is re-fetched.
To address this, we can factor the Iris dataset into its own asset.
↪   This will allow us to:
Use the asset as input to additional notebooks.
This means all notebooks analyzing the Iris dataset will use the same
↪   source data, which we only have to fetch once.
Materialize notebooks without fetching data for each materialization.
Instead of making potentially expensive API calls, Dagster can fetch
↪   the data from the previous materialization of the Iris dataset
↪   and provide that data as input to the notebook.
```

**Markdown Syntax Format**

We also retrieve relevant documentation from the web to help you with
↪   the task:

Documentation Source:
release-1-7-2.dagster.dagster-docs.io/integrations/dagstermill/using-
notebooks-with-dagster.md

Documentation Title:
Using Jupyter notebooks with Papermill and Dagster Tutorial

Documentation Content:
When clicked, Dagster will render the notebook - referenced in the
↪   `notebook_path`parameter - that'll be executed when the
↪   `iris_kmeans_jupyter`asset is materialized:

!Click the **Materialize**button. To view the execution as it happens,
↪   click the **View**button in the alert that displays.

After the run completes successfully, you can view the executed
↪   notebook in the UI. Click the asset again and locate the **View
↪   Notebook**button in the **Materialization in Last Run**section of
↪   the sidebar:

!Click the button to display the executed notebook - specifically,
↪   the notebook that was executed and written to a persistent
↪   location:

!Step 5: Add an upstream asset#
-----------------------------

While our `iris-kmeans`notebook asset now materializes successfully,
↪   there are still some improvements we can make. The beginning of
↪   the notebook fetches the Iris dataset, which means that every
↪   time the notebook is materialized, the data is re-fetched.

To address this, we can factor the Iris dataset into its own asset.
↪   This will allow us to:

**Use the asset as input to additional notebooks.**This means all
↪   notebooks analyzing the Iris dataset will use the same source
↪   data, which only have to fetch once.

**Materialize notebooks without fetching data for each
↪   materialization.**Instead of making potentially expensive API
↪   calls, Dagster can fetch the data from the previous
↪   materialization of the Iris dataset and provide that data as
↪   input to the notebook.

In this step, you'll:

Create the Iris dataset assetProvide the Iris dataset as input to the
↪   notebookModify the notebook

**Simplified HTML Format**

```
We also retrieve relevant documentation from the web to help you with
↪  the task:

Documentation Source:
release-1-7-2.dagster.dagster-docs.io/integrations/dagstermill/using-
notebooks-with-dagster.html

Documentation Title:
Using Jupyter notebooks with Papermill and Dagster Tutorial

Documentation Content:
If you execute these cells, several plots of the Iris dataset will be
↪  created:
<p>Next, we conduct our K-means analysis:</p>
estimator
↪  =sklearn.cluster.KMeans
(n_clusters=3)
estimator.fit(iris[
["Sepal length (cm)",
"Sepal width (cm)",
"Petal length (cm)",
"Petal width (cm)"
]])

<p>Lastly, we plot the results of the K-means analysis. From the
↪  plots, we can see that one species of Iris is separable from the
↪  other two, but a more sophisticated model will be required to
↪  distinguish the other two species:</p>
<p>Like many notebooks, this example does some fairly sophisticated
↪  work, including producing diagnostic plots and a statistical
↪  model. For now, this work is locked away in the
↪  .ipynbformat, only reproducible using a complex
↪  Jupyter setup, and only programmatically accessible within the
↪  notebook context. We'll address this in the remainder of the
↪  tutorial.</p>
<h2>Step 2: Create a Dagster asset from the Jupyter
↪  Notebook#</h2>
<p>By creating a Dagster asset from our notebook, we can integrate
↪  the notebook as part of our data platform. This enables us to
↪  make its contents more accessible to developers, stakeholders,
↪  and other assets in Dagster.</p>
<p>To create a Dagster asset from a Jupyter notebook, we can use the
↪  define_dagstermill_assetfunction.
```

# J  Datasheet for Spider2-V

## J.1  Motivation

- **For what purpose was the dataset created?** Was there a specific task in mind? Was there a specific gap that needed to be filled? Please provide a description.

  The proposal of Spider2-V intends to investigate the capability of existing large language model (LLM) or vision language model (VLM) based agents in automating real-world data science and engineering workflows. Task examples in Spider2-V establish three novel challenges for LLM/VLM-based data agents: 1) a thorough analysis across the entire data pipeline beyond traditional data processing and querying tasks, 2) the integration of

enterprise-level professional applications, and 3) the incorporation of intensive GUI controls over the web pages with classic coding actions for these specialized software.

- **Who created the dataset (*e.g.*, which team, research group) and on behalf of which entity (*e.g.*, company, institution, organization)?**

  Ruisheng Cao, Fangyu Lei, Haoyuan Wu, Jixuan Chen, Yeqiao Fu, Hongcheng Gao, Xinzhuang Xiong, Hanchong Zhang, Yuchen Mao, Wenjing Hu, Tianbao Xie, Hongsheng Xu, Danyang Zhang, Sida Wang, Ruoxi Sun, Pengcheng Yin, Caiming Xiong, Ansong Ni, Qian Liu, Victor Zhong, Lu Chen, Kai Yu, and Tao Yu, on behalf of XLang Lab of the University of Hong Kong [12], X-LANCE Lab of SJTU AI Institute, Google Cloud AI Research, Google DeepMind, Salesforce Research, Yale University, Sea AI Lab, and University of Waterloo create the environment and the task set.

## J.2 Composition

- **What do the instances that comprise the dataset represent (*e.g.*, documents, photos, people, countries)?** Are there multiple types of instances (*e.g.*, movies, users, and ratings; people and interactions between them; nodes and edges)? Please provide a description.

  Each instance in Spider2-V represents a real-world data science and engineering task which can be fulfilled by data scientists on a computer. These tasks cover the entire data pipeline, including 1) data warehousing, 2) data ingestion and integration, 3) data transformation, 4) data analysis and visualization, 5) traditional data processing, 6) data orchestration, and 7) IT service management.

- **How many instances are there in total (of each type, if appropriate)?**

  We totally instantiate 494 tasks spanning across 20 professional applications and covering the entire data pipeline. We also provide a document warehouse containing 11, 231 documents which are crawled and pre-processed from the official websites of these professional data science and engineering software to support retrieval augmented agent framework.

- **Does the dataset contain all possible instances or is it a sample (not necessarily random) of instances from a larger set?** If the dataset is a sample, then what is the larger set? Is the sample representative of the larger set (*e.g.*, geographic coverage)? If so, please describe how this representativeness was validated/verified. If it is not representative of the larger set, please describe why not (*e.g.*, to cover a more diverse range of instances, because instances were withheld or unavailable).

  The entire task set contains 494 real-world tasks which stem from the following 3 sources: 1) dataset SheetCopilot [18] from which we randomly sample 31 tasks, 2) benchmark WorkArena [6] from which we sample one case for each distinct task type, and 3) official tutorials or guides on professional applications (e.g., dbt, Airflow, Dagster, Superset, etc.). During construction, we focus on the task diversity and reality. And each task sample goes through strict manual cross-validation by at least two annotators (see § 3.1).

- **What data does each instance consist of?** "Raw" data (*e.g.*, unprocessed text or images) or features? In either case, please provide a description.

  Each instance in Spider2-V includes 1) a text-format natural language instruction (task goal), 2) two lists of functions defining how to set up the initial environment and how to evaluate the final outcome respectively, as well as 3) some other task metadata. All information is organized in the JSON format (see appendix G). They can be directly loaded by the provided environment and support interactive and continuous communication with data agents until task completion. All types of auxiliary media (*e.g.*, files, images, archives or scripts associated with different tasks) are provided in company with the raw JSON data.

- **Is there a label or target associated with each instance?** If so, please provide a description.

  Yes. For each instance, it includes a task goal, and one evaluation script methods to determine whether the current task is completed successfully.

- **Is any information missing from individual instances?** If so, please provide a description, explaining why this information is missing (*e.g.*, because it was unavailable). This does not include intentionally removed information, but might include, *e.g.*, redacted text.

---

[12]https://www.xlang.ai/

For part of the task instances, we require the user to provide a personal account of specific professional applications (*e.g.*, `BigQuery` and `Snowflake`), see appendix G for details. We can not provide a public account. The reason is that, if multiple users share one trial account, the situation may occur that when one person just finished the task using his/her data agent and prepare to evaluate the results, another person happens to reset the environment for the same task, thus destroying all finished states in the cloud workspace of this real account. However, we do provide templates and instructions about how to register and use these real accounts for Spider2-V.

- **Are there any errors, sources of noise, or redundancies in the dataset?** If so, please provide a description.

  Currently, no. We try to eliminate errors and noise via strict cross-validation by at least two annotators. Versions of different software or libraries are fixed for robustness and consistency. Redundancies should not exist since each task sample is uniquely selected for diversity. And we will keep the questions and contact channels open to address any potential annotation errors and software update issues.

- **Is the dataset self-contained, or does it link to or otherwise rely on external resources (*e.g.*, websites, tweets, other datasets)?** If it links to or relies on external resources, a) are there guarantees that they will exist, and remain constant, over time; b) are there official archival versions of the complete dataset (*i.e.*, including the external resources as they existed at the time the dataset was created); c) are there any restrictions (*e.g.*, licenses, fees) associated with any of the external resources that might apply to a dataset consumer? Please provide descriptions of all external resources and any restrictions associated with them, as well as links or other access points, as appropriate.

  Spider2-V includes a subset of two existing benchmarks, namely SheetCopilot [18] and WorkArena [6]. In response to sub-questions: a) Data has been downloaded locally and transformed to the target format of Spider2-V. b) Yes. Spider2-V archives all the annotations. c) These two benchmarks are released under GPL-3.0 license and Apache-2.0 license respectively. Therefore, we have access to them and do not violate any protocol.

- **Does the dataset contain data that might be considered confidential (*e.g.*, data that is protected by legal privilege or by doctor–patient confidentiality, data that includes the content of individuals' non-public communications)?** If so, please provide a description.

  No.

- **Does the dataset contain data that, if viewed directly, might be offensive, insulting, threatening, or might otherwise cause anxiety?** If so, please describe why.

  No.

- **Does the dataset identify any subpopulations (*e.g.*, by age, gender)?** If so, please describe how these subpopulations are identified and provide a description of their respective distributions within the dataset.

  No.

- **Is it possible to identify individuals (*i.e.*, one or more natural persons), either directly or indirectly (*i.e.*, in combination with other data) from the dataset?** If so, please describe how.

  No. Only the research group of Spider2-V may be identified, because we use the benchmark name in some task instances (*e.g.*, file name or folder called "`Spider2`") for demonstration use case. For other individuals, it is impossible since we only use anonymous or fabricated names.

- **Does the dataset contain data that might be considered sensitive in any way (*e.g.*, data that reveals race or ethnic origins, sexual orientations, religious beliefs, political opinions or union memberships, or locations; financial or health data; biometric or genetic data; forms of government identification, such as social security numbers; criminal history)?** If so, please provide a description.

  No.

### J.3 Collection Process

- **How was the data associated with each instance acquired?** Was the data directly observable (*e.g.*, raw text, movie ratings), reported by subjects (*e.g.*, survey responses), or indirectly inferred/derived from other data (*e.g.*, part-of-speech tags, model-based guesses for age or language)? If the data was reported by subjects or indirectly inferred/derived from other data, was the data validated/verified? If so, please describe how.

  All data and information used in Spider2-V are directly observable from 1) existing benchmarks (SheetCopilot and WorkArena) and 2) websites of official tutorials for various professional applications. Indeed, the JSON-format metadata of each task contains one field "`source`" indicating where the data comes from (see Listing 1).

- **What mechanisms or procedures were used to collect the data (*e.g.*, hardware apparatuses or sensors, manual human curation, software programs, software APIs)?** How were these mechanisms or procedures validated?

  Spider2-V is developed with a virtual machine based computer desktop. And tasks are annotated through manual human curation and manual cross validation on the virtual machine. The detailed procedure (including the verification step) is summarized and demonstrated with one concrete example in § 3.1.

- **If the dataset is a sample from a larger set, what was the sampling strategy (*e.g.*, deterministic, probabilistic with specific sampling probabilities)?**

  For the subset that stems from existing benchmarks, we adopt uniform sampling for different task types to ensure diversity. For the main part which originates from official tutorials of various professional applications, we adopt breadth-first-search to collect diverse topics (removing duplicates). Then, we iterate through these tutorials and filter those tutorials that can not be instantiated.

- **Who was involved in the data collection process (*e.g.*, students, crowdworkers, contractors) and how were they compensated (*e.g.*, how much were crowdworkers paid)?**

  All the development of the environment and task set are completed by the authors.

- **Over what timeframe was the data collected?** Does this timeframe match the creation timeframe of the data associated with the instances (*e.g.*, recent crawl of old news articles)? If not, please describe the timeframe in which the data associated with the instances was created.

  The task set is collected from official tutorials of professional applications in Table 5 from Jan. 20th, 2024 to Jun. 2nd, 2024.

- **Did you collect the data from the individuals in question directly, or obtain it via third parties or other sources (e.g., websites)?**

  Data is collected from official websites of various professional applications. For the complete checklist of all websites, see Table 8.

- **Were the individuals in question notified about the data collection?** If so, please describe (or show with screenshots or other information) how notice was provided, and provide a link or other access point to, or otherwise reproduce, the exact language of the notification itself.

  Not applicable. We do not collect data from individuals.

- **Did the individuals in question consent to the collection and use of their data?** If so, please describe (or show with screenshots or other information) how consent was requested and provided, and provide a link or other access point to, or otherwise reproduce, the exact language to which the individuals consented.

  Not applicable. We do not collect data from individuals.

- **If consent was obtained, were the consenting individuals provided with a mechanism to revoke their consent in the future or for certain uses?** If so, please provide a description, as well as a link or other access point to the mechanism (if appropriate).

  Not applicable. We do not collect data from individuals.

### J.4 Uses

- **Has the dataset been used for any tasks already?** If so, please provide a description.

Data agents based on different LLMs and VLMs have been evaluated on Spider2-V in this paper. And the detailed configurations are elaborated in the Experiment section (§ 4.1).

- **Is there a repository that links to any or all papers or systems that use the dataset?** If so, please provide a link or other access point.

  Yes. We construct a project website (`https://spider2-v.github.io/`) tracking the latest paper, code repository, task data, virtual machine snapshots, crawled documents and results for Spider2-V.

- **Is there anything about the composition of the dataset or the way it was collected and preprocessed/cleaned/labeled that might impact future uses?** For example, is there anything that a dataset consumer might need to know to avoid uses that could result in unfair treatment of individuals or groups (*e.g.*, stereotyping, quality of service issues) or other risks or harms (*e.g.*, legal risks, financial harms)? If so, please provide a description. Is there anything a dataset consumer could do to mitigate these risks or harms?

  No.

## J.5  Distribution

- **Will the dataset be distributed to third parties outside of the entity (*e.g.*, company, institution, organization) on behalf of which the dataset was created?** If so, please provide a description.

  Yes. Both the environment, the task set and experiment code have been open-sourced in Github `https://github.com/xlang-ai/Spider2-V`.

- **How will the dataset will be distributed (*e.g.*, tarball on website, API, GitHub)?** Does the dataset have a digital object identifier (DOI)?

  The platform is open-sourced in GitHub `https://github.com/xlang-ai/Spider2-V`. The task set is also released at Hugging Face `https://huggingface.co/datasets/xlangai/ubuntu_spider2v`. We do not apply for a DOI.

- **When will the dataset be distributed?**

  Both the executable environment and the task set have already been made public on 16 July, 2024.

- **Will the dataset be distributed under a copyright or other intellectual property (IP) license, and/or under applicable terms of use (ToU)?** If so, please describe this license and/or ToU, and provide a link or other access point to, or otherwise reproduce, any relevant licensing terms or ToU, as well as any fees associated with these restrictions.

  Spider2-V is open-sourced under Apached-2.0 license.

- **Have any third parties imposed IP-based or other restrictions on the data associated with the instances?** If so, please describe these restrictions, and provide a link or other access point to, or otherwise reproduce, any relevant licensing terms, as well as any fees associated with these restrictions.

  Yes. IPs in some countries do not have access to the service delivered by some professional applications (*e.g.*, `BigQuery` and `Snowflake`) involved in Spider2-V. However, we do provide the solution to resolve this issue by setting the network proxy. And this constrained task subset is marked as a separate split only for interested users.

- **Do any export controls or other regulatory restrictions apply to the dataset or to individual instances?** If so, please describe these restrictions, and provide a link or other access point to, or otherwise reproduce, any supporting documentation.

  No.

## J.6  Maintenance

- **Who will be supporting/hosting/maintaining the dataset?**

  The authors, on behalf of XLang Lab of the University of Hong Kong, will support, host, and maintain both the environment and the task set of Spider2-V.

- **How can the owner/curator/manager of the dataset be contacted (*e.g.*, email address)?**

Issues and discussions on GitHub and Hugging Face are welcome. Benchmark users can also seek help from Ruisheng Cao (`ruishengcao@gmail.com`), and Tao Yu (`tao.yu.nlp@gmail.com`).

- **Is there an erratum?** If so, please provide a link or other access point.

  Currently, no. Errata will be anounced if there is any in future.

- **Will the dataset be updated (*e.g.*, to correct labeling errors, add new instances, delete instances)?** If so, please describe how often, by whom, and how updates will be communicated to dataset consumers (*e.g.*, mailing list, GitHub)?

  Spider2-V will be continuously developed and maintained. Updates will be released on GitHub and project website from time to time. Errata may be released to correct errors if needed in future.

- **If the dataset relates to people, are there applicable limits on the retention of the data associated with the instances (*e.g.*, were the individuals in question told that their data would be retained for a fixed period of time and then deleted)?** If so, please describe these limits and explain how they will be enforced.

  No.

- **Will older versions of the dataset continue to be supported/hosted/maintained?** If so, please describe how. If not, please describe how its obsolescence will be communicated to dataset consumers.

  Yes. Old versions of Spider2-V (including the environment code, task examples, and virtual machine snapshots) can be accessed through GitHub and Hugging Face.

- **If others want to extend/augment/build on/contribute to the dataset, is there a mechanism for them to do so?** If so, please provide a description. Will these contributions be validated/verified? If so, please describe how. If not, why not? Is there a process for communicating/distributing these contributions to dataset consumers? If so, please provide a description.

  We sincerely welcome that any benchmark user contributes features or report bugs for Spider2-V through the mechanisms like pull request, issues, *etc*. on GitHub or directly e-mail the authors. If new environments or task sets are crafted and prepared, it is also welcome that the creators notify us through the contact channels mentioned before, such that we can update the indices of available task sets.

