# OpenReview forum: "Spider2-V: How Far Are Multimodal Agents From Automating Data Science and Engineering Workflows?"
_NeurIPS.cc/2024/Datasets_and_Benchmarks_Track — NeurIPS 2024 Track Datasets and Benchmarks Spotlight_

### Official Review · Reviewer_Lwbw · 2024-07-21
**Rebuttal update: Accept**

**Rating:** 6
**Confidence:** 2
**Correctness:** Yes.
**Clarity:** See comments above.

**Review:**

**Rebuttal update:**

My concerns have been addressed therefore I am updating my ranking from 3 to 6.

**Old review:**

The authors claim in the paper checklist that they have included limitations of their work as well as societal impact but none of these sections could be found in the main paper nor the supplementary material. Based on the Neurips guidelines this paper can not be accepted to the conference in its current form.

It is also mentioned in the checklist that the authors have explained amount of training resources (e.g. number of gpus etc) used in their experiments while I could not find any of those details in the paper.

Here are my additional comments on the paper:

The subject investigated by the paper is interesting in context of workflow automation agents and is definitely a subject of interest in the community. However, as a new benchmark/dataset the evaluation of models needs to be much more rigorous. For instance, none of the tables or figures include any confidence bounds or error bars. Moreover, the results are not presented in a way that would make it easy to compare performance of known baseline models on different benchmark datasets (like reference 11 "SWE-Bench") versus spider2-V. The same holds for tables 4 and 5 which only include results from GPT-4o. In my opinion including at least one other baseline in each table (other benchmark data sets for table 3 and other benchmark models for tables 4 and 5 could be beneficial. Also the flow of the paper seems odd to me specially since the "related work" section is at the end of the paper instead of the beginning.

I am also a bit confused on the novelty claim of the paper (since there are so many hints on novelty of tasks etc instead of explicit listing of the contributions of the paper in the introduction. Is the main motivation of the work the fact that "... there is currently no benchmark that integrates both code generation and GUI controls for data science and engineering."? Can you please elaborate on the contribution of the paper beyond expanding to new tasks? Table2 shows several other benchmarks that support gui+ control/exec/env, is your contribution comparing to them (for example OSWorld, WorkArena, and WebArena) the set of new tasks defined for engineering and data science ?

**Strengths:**

Developing benchmarks and datasets for automation tasks is an important topic for development of new agents and this paper provides an in depth explanation of steps involved in design of such benchmark in context of several complex engineering and data science tasks. With high-level comparison with several relevant benchmarks. The writing of the paper is clear although its structure could be improved.

**Additional Feedback:**

NA

**Documentation:**

Yes

**Limitations:**

No. This paper does not include a limitations section nor does it discuss it in the conclusion. For some suggestions on what to improve please see my comments above.

**Opportunities For Improvement:**

Please see the review section above.

**Relation To Prior Work:**

No. See comments above.

**Summary And Contributions:**

This paper is about a new multimodal benchmark for data science and engineering work flow automation. The proposed new benchmark includes several multi-step tasks with specific advantage of supporting "enterprise service" and "gui support" in addition to a controllable and executable environment in context of engineering and data science tasks. The paper includes detailed explanation of steps involved in each task and their importance in addition to benchmarking of GPT-4o on different tasks and subsets of the dataset.

---

> ### Author Rebuttal · Authors · 2024-08-20
>
> We thank the reviewer for their time and attention to our work.
>
> **Q1: Missing limitation and social impact section.**
>
> **A1:** We have mentioned the limitations and social impact in our main paper.
> - The **limitations** include: a) The interaction efficiency is unsatisfactory; b) The usage of real accounts may encounter real-world problems; c) The annotation work is not scalable and requires domain expertise.
> - The **social impacts** include: 1) enhance productivity for data scientists and engineers; 2) revolutionize data science and engineering workflow automation.
>
> These claims are all mentioned in the main text. Here is the evidence for each point:
>
> **1) For Limitation:**
>
> **a) The interaction efficiency is unsatisfactory.** The statements are:
> - in line 260: “_However, due to length constraints and considerations of cost-effectiveness, we are unable to further enlarge the length of the history trajectories. This also points out that interaction efficiency is a very serious issue and research direction_”.
> - in the Conclusion section: “_Future work includes … , and 3) improving the interaction efficiency_”.
> - in the prompt template (Appendix E.1.2), we listed that “_For time efficiency, you can return one line or multiple lines of python code to perform continuous actions in one response_” and “_For efficiency, you CAN predict multiple actions in one response_”.
>
> These sentences all emphasize the limitation of Spider2-V regarding interaction efficiency.
>
> **b) The usage of real accounts may encounter real-world problems.** Here is the evidence:
> - in line 206-208: “... _mostly involve real accounts (BigQuery and Snowflake). Compared to other tasks which can be accomplished in a local host, these dynamic real-world scenarios incur extra burden on data agents, such as network connection delay and pop-up windows_”.
> - in line 236-237: “... _tasks involving real accounts, which can be partly explained by various complex situations and accidents in the real world (e.g., the web server suddenly crashes)_”.
>
> These statements demonstrate that when evaluating tasks involving authentic user accounts in Spider2-V, we could encounter failures caused by remote cloud-hosted service providers.
>
> **c) The annotation work is not scalable and requires domain expertise.** The evidence is:
> - line 54-56: “_10 authors with computer science backgrounds developed 170 automatic task setup configurations and 151 customized evaluation metrics_”.
> - line 145: “_On average, the annotation of one task (including cross-validation) costs roughly 4 hours_”.
> - section 2.1: “_Five common operations to set up the initial environment_” requires the annotator to “_invoke tool-specific API calls_”, “_execute a shell script_”, and “_run web browser simulation with Playwright_”
> - section 2.2: “_Three generic methods for task evaluation_” states that annotators need to “_write task-dependent functions to retrieve the desired result from the open-ended resulting state_”.
> - section 3.1: The complicated annotation pipeline involves “_Collect tutorials_”, “_Learn tutorials_”, “_Write instructions_”, “_Write environment setup functions_”, “_Write task-specific evaluation functions_”, and “_Cross-validate on VM_”.
>
> All the text content above consolidates that the annotation of Spider2-V prevents large-scale crowd-sourcing among non-experts.
>
> **2) For Social Impact:** The social impact is mentioned in the Abstract and Introduction sections. Concretely,
> - in the Abstract:
>     - line 4-7: “... _With the rapid progress of VLMs in multimodal understanding and code generation, VLM-based agents have the potential to automate these workflows, enhancing productivity for data scientists and engineers while democratizing large data access_”.
>     - line 21-22: “_paves the way for practical multimodal agents to revolutionize data science and engineering workflow automation_”.
> - in the Introduction:
>     - line 29-31: “_LLM/VLM-based autonomous agents have the potential to automate these workflows, enhancing productivity for data scientists and engineers while democratizing access to large data. This has attracted increasing attention in both academia and industry_”.
>     - line 61-63: “... _lays the groundwork for developing practical multimodal agents that can revolutionize the automation of data science and engineering workflows_”.
>
> And in each section above, we describe two points, that is a) enhancing productivity, and b) automating data science and engineering workflows.
>
> While **the NeurIPS Paper Checklist Guidelines[17] does not require a separate limitation and social impact section (see below)**, we will supplement them by combining existing content from the manuscript to improve readability.
> - **2. Limitations:** “_The authors are encouraged to create a separate "Limitations" section in their paper_”. Thus, a separate Limitation section is **not compulsory**.
> - **10. Broader Impacts:** “_If appropriate for the scope and focus of your paper, did you discuss potential negative societal impacts of your work?_” Accordingly, the discussion of potential social impact is **optional**.
>
> And we would like to thank the reviewer for pointing out this issue. The following two sections will be added into the Appendix.

---

> > ### Author Rebuttal · Authors · 2024-08-20
> >
> > **1) Limitation Section:**
> > ```txt
> > Limitation
> >
> > Despite the contribution of Spider2-V, there are still several limitations: a) One concern is about the scalability of tasks. The complicated annotation procedure (elaborated in sections 2.2, 2.3, 3.1) requires professional expertise and tool-specific programming skills. Annotators need to understand how to manipulate the professional applications, manually design the task, and write instructions, environment setup methods and evaluation functions. This may prevent large-scale crowd-sourcing among non-experts. b) Another major issue is the interaction efficiency. To resolve real-world tasks in Spider2-V, LLM agents need multiple turns of interaction with the executable environment. During the interaction process, the prompt length and API cost increase dramatically if we want to insert more history context. However, we hope this arising problem can inspire more interesting work in improving the agent interaction efficiency. c) Lastly, the evaluation on tasks involving authentic user accounts in Spider2-V may encounter failures due to unexpected errors caused by the remote cloud-hosted enterprise, e.g., the web server suddenly crashes.
> > ```
> >
> > **2) Social Impact Section:**
> > ```text
> > Social Impact
> >
> > This paper proposes Spider2-V, the first-of-its-kind multimodal agent benchmark focusing on professional data science and engineering workflows, featuring 494 real-world tasks in authentic computer environments and 20 enterprise-level professional applications. The goal is to enhance productivity for data scientists and lay the groundwork for developing practical multimodal agents that can revolutionize the automation of data science and engineering workflows. The potential negative impact is that, once more effective agents are delivered for this type of task set, it could lead to some degree of unemployment among data scientists and engineers. Besides, for tasks involving authentic user accounts, benchmark users need to create authentic accounts to exploit the cloud-hosted sandbox services and test their agents. It potentially increases the management burden for software service providers. Furthermore, since the interaction proceeds in a fully realistic environment, the developed universal digital agent could be used to bypass CAPTCHA systems in the future.
> > ```
> > ----
> >
> > **Q2: The author did not mention the amount of training resources (e.g. number of gpus etc).**
> >
> > **A2:** We claim in the checklist that “_Did you include the total amount of compute and the type of resources used? [Yes]_”. This is because:
> >
> > - We **did not train a specialized agent model and do not need GPUs**, much less an internal cluster.
> > - We directly invoke remote APIs of those cloud providers which can even be performed on a personal laptop.
> >
> > Thus, we do not mention the number of GPUs. To further alleviate your concerns regarding experimental resources, we will elaborate on the following aspects:
> >
> > **1) Training resources:** For more clarity, we will add the following statements in the section “4.1 Experiment Settings -> LLMs and VLMs”.
> >
> > ```txt
> > ... Concretely, for closed-source models, we directly resort to their API services with versions gpt-4o-2024-05-13, gpt-4-1106-vision-preview, gemini-1.5-pro-latest and claude-3-opus-20240229. While for open-weight models, we use the GroqCloudTM API platform[12] where the hardware resources are provided by Groq.
> > ```
> >
> > **2) Average running time and cost per task:** Note that the running time complexity is tightly connected to the maximum interaction turns, trajectory window size, and network delay when calling APIs.
> >
> > | Model | Expected Running Time | Average Prompt Tokens | Average Completion Tokens | Average Cost Per Task |
> > | ----------- | ------------------------ | --------------- | ------------ | ------------ |
> > | GPT-4o  | 8.7 minutes | 0.32M        | 1.6K        | $1.64      |
> >
> > On average, each task costs $1.64 with gpt-4o-2024-05-13. The detailed configuration is (also the most powerful agent baseline in our main experiment):
> > - action_space = pyautogui code
> > - observation_space = SoM
> > - prompt tokens include image pixels
> > - max_turn = 15
> > - trajectory_window_size = 3
> > - w/ retrieval augmentation (top_k = 4, chunk_size = 512 tokens)
> >
> > **3) Real-time environment delay:** The expected running time for each task can be further decoupled into:
> > 1. environment setup time;
> > 2. response time when calling APIs or querying LLMs (may depend on the network traffic);
> > 3. time delay to perform each action in the computer desktop (this is controllable, and we set the time delay to 1 second for the action to take effect);
> > 4. time delay to obtain the screenshot and accessibility tree (observations) after each action grounding;
> > 5. evaluation time at the end of the episode.
> >
> > | Environment Setup  | Per Action Time Delay | Per Observation Time Delay | Evaluation Time |
> > | ----------- | ----------- | ----------- | ----------- |
> > | ~ 1 minute   | 1 second               | ~ 2 seconds        | < 1 minute          |
> >
> > According to the table above, we can find that the majority time is spent on multi-turn API calls due to long prompt tokens and slow inference time. Interaction efficiency is a promising future direction.
> >
> > The statistics above will all be added in the Appendix.
> >
> > ----
> >
> > **Q3: No confidence bounds or error bars in experiment results.**
> >
> > **A3:** As we claimed in the checklist “_Did you report error bars?_”, the answer is "_[N/A]_". This is because:
> > - When invoking APIs of closed-source models, the cost is relatively high ($1.64 per task with GPT-4o, presented in Q2). We are on a limited budget and cannot afford to run duplicated experiments and report error bars.
> > - For open-sourced models, the performances are still very poor (success rates are less than 2% according to Table 3), making it less meaningful to study the standard error.
> >
> > Therefore, we did not include the error bars.

---

> > > ### Author Rebuttal · Authors · 2024-08-20
> > >
> > > **Q4: Did not compare performance of known baseline models on different benchmarks, e.g., results on other benchmarks in Table 3 and other agent baselines in Table 4 or 5.**
> > >
> > > **A4:** We will address your concerns about other benchmarks and agent baselines.
> > >
> > > **1) For other benchmarks:** One trend of latest benchmarks is to provide an executable and interactive environment. And different dynamic benchmarks (WebArena[1], OSWorld[6], AndroidWorld[8]) focus on **a) different platforms and downstream fields**, and support **b) different environments and action/observation spaces**. Some advanced agent methods are even specially devised for a narrow task set (discussed later). Therefore, **performances on these benchmarks with interactive environments are not comparable**, unlike previous literature which focuses on a specific downstream task or topic (e.g., question answering).
> > >
> > > In the table below, we provide the overall results of different benchmarks equipped with an executable environment using GPT-4/GPT-4V to validate that Spider2-V is very challenging. Notice that, we exclude SWE-bench[3] and MLAgentBench[11], because they do not use the task success rate as the evaluation metric.
> > >
> > > | Benchmark | Model | Overall Success Rate |
> > > | ---- | ---- | ---- |
> > > | WebArena[1]      |  GPT-4    | 14.4    |
> > > | VisualWebArena[16] | GPT-4V | 16.4  |
> > > | OSWorld[6]        |  GPT-4V  |  12.2   |
> > > | AndroidWorld[7] |  GPT-4V  |  30.6   |
> > > | WorkArena[15]  |  GPT-4V  |   41.8   |
> > > | Spider2-V          |  GPT-4V  |   14.0   |
> > >
> > > **2) As for different agent baselines in Table 4 and 5:** The motivation of Table 4 and 5 is to offer comprehensive results on the benchmark and environment itself, including:
> > > - performances on different task subsets splitted by different criteria;
> > > - ablation study on different action space and observation types;
> > > - decomposition of several agent techniques in our best-performing method.
> > >
> > > Thus, **it is sufficient to use the same agent baseline via controlling variables**. To alleviate your concerns about the rigor of the conclusion, we supplement the ablation Table 4 with another model GPT-4V.
> > >
> > > | Task Splits  | Ratio (%)   |  GPT-4o (%)  |  GPT-4V (%)  |
> > > | ---- | ---- | ---- | ---- |
> > > | Easy      | 19.8  |  **38.8**            |       **35.7**   |
> > > | Medium | 62.8  | 9.7                     |      10.6     |
> > > | Hard      | 17.4  | 1.2                     |    0.0         |
> > > | w/o account |  66.0  | **15.6**        |   **15.4**    |
> > > | w/ account   |  34.0  | 10.6              |      11.2       |
> > > | CLI               | 5.7     |  7.1               |       8.5             |
> > > | GUI              | 37.7   |  **20.1**        |      **21.2**       |
> > > | CIL+GUI      | 56.7   | 10.6               |    9.7      |
> > > | Abstract       | 50      | 11.3               |    11.3      |
> > > | Verbose       | 50      | **16.2**         |   **16.6** |
> > >
> > > The results above are based on the entire benchmark instead of a task subset this time. Accordingly, the conclusions regarding difficulty levels, authentic user accounts, interaction interfaces, and instruction types remain unchanged.
> > >
> > > Besides, previous work which proposes a complicated agent framework mostly focuses on a narrow downstream task or platform. We can only extract the core ideas in these agent frameworks, adapt them to our interactive environment and tasks, and experiment with these variants or methods. And we have verified the following three methods:
> > >
> > > **a) Set-of-Mark:** Firstly proposed by SoM[13], and further adopted by WebArena[1] and OSWorld[6]. The implementation details are illustrated in Supplementary material E3.1.
> > >
> > > **b) Execution Feedback:** Adopted in Reflexion[14] to include the execution results. In our case, we utilize the execution error messages after action execution, exemplified in Appendix C.3 Figure 9.(3).
> > >
> > > **c) Retrieval-augmented Context:** This is a widely used technique[3] regarding knowledge-intensive tasks. And the prompt template and use case are demonstrated in Appendix E.3.
> > >
> > > The ablation study of different agent baselines is presented in the bottom part of Table 5, which demonstrates the effectiveness of each method.
> > >
> > > | Agent Method | Success Rate (%) |
> > > | ---- | ---- |
> > > | No advanced methods | 11.4 |
> > > | w/ Set-of-Mark | 15.6 |
> > > | w/ Execution Feedback | 13.6 |
> > > | w/ Retrieval-augmented Context | 14.4 |
> > > | w/ all three methods above     | **16.3** |
> > >
> > > ----
> > >
> > > **Q5: The paper structure is odd. Why is the Related Work section at the end instead of at the beginning?**
> > >
> > > **A5:** This structure is also feasible because many published excellent benchmark papers (e.g., WebArena[1], MineDojo[2], and SWE-bench[3]) also make this choice. I think the underlying motivation is to enable readers to quickly understand the contributions of this article and check the details.

---

> > ### Author Rebuttal · Authors · 2024-08-20
> >
> > **Q6: Confusion about the novelty: is the contribution merely proposing a new task set in the data science and engineering field?**
> >
> > **A6:** We are deeply sorry that the novelty of Spider2-V was not made clear. We will elaborate the core contributions and compare with the works you mentioned.
> >
> > **1) Contributions and novelty:** The itemized contributions below will be enumerated at the end of the Introduction section. Thanks for your suggestion.
> >
> > - We propose a new benchmark Spider2-V, which contains $494$ real-world tasks across **the complete data science and engineering workflows** (from data warehousing to orchestration), instead of the single data processing stage.
> > - We incorporate 20 **enterprise-level professional applications** (e.g., BigQuery, dbt, Airbyte, etc.), some of which even need **authentic user accounts** to bridge the gap between academia and industry.
> > - We are the first to **integrate both command line (CLI) and graphical user interfaces (GUI)**, to support intensive GUI controls in the data-related or coding field.
> > - We also provide an interactive executable computer environment with a crawl-and-preprocessed **document warehouse** for agent retrieval.
> >
> > **2) Benchmark comparison:** In line 165-175, we compare Spider2-V with other benchmarks (also presented in Table 2). Spider2-V differentiates with them in the following aspects:
> >
> > - **an interactive executable environment.** Instead of providing static input-output pairs (like Spider[4], DS-1000[5], Arcade[8], and Mind2web[10]), Spider2-V is equipped with a dynamic computer desktop such that agents can proactively explore it;
> > - **intensive GUI operations apart from coding.** Unlike traditional coding or data science domains (like SWE-bench[3], SheetCopilot[9], and MLAgentBench[11]), experienced data scientists frequently manipulate the UIs of professional software to simplify the data workflow (e.g., enabling a specific function on the UI page or visualizing the graph view of data inputs after coding). We present a more practical scenario regarding how programmers work in the real world;
> > - **enterprise-level professional software.** Compared to prevalent GUI-enabled environments (like WebArena[1], OSWorld[6], and AndroidWorld[7]) on different platforms, we focus on enterprise-level professional applications instead of daily life software (e.g., e-mail, calendar, and alarm clock). These applications require domain expertise to manipulate and even rely on cloud-hosted remote services. We also crawl and clean raw documents of involved professional software to support retrieval-augmented agent framework.
> >
> > The paragraph in section 3.3 will be replaced with the items above for clarity.
> >
> > ----
> >
> > **References:**
> >
> > - [1] Zhou, Shuyan, et al. "WebArena: A Realistic Web Environment for Building Autonomous Agents." The Twelfth International Conference on Learning Representations.
> > - [2] Fan, Linxi, et al. "Minedojo: Building open-ended embodied agents with internet-scale knowledge." Advances in Neural Information Processing Systems 35 (2022): 18343-18362.
> > - [3] Jimenez, Carlos E., et al. "SWE-bench: Can Language Models Resolve Real-world Github Issues?." The Twelfth International Conference on Learning Representations.
> > - [4] Yu, Tao, et al. "Spider: A Large-Scale Human-Labeled Dataset for Complex and Cross-Domain Semantic Parsing and Text-to-SQL Task." Proceedings of the 2018 Conference on Empirical Methods in Natural Language Processing. Association for Computational Linguistics, 2018.
> > - [5] Lai, Yuhang, et al. "DS-1000: A natural and reliable benchmark for data science code generation." International Conference on Machine Learning. PMLR, 2023.
> > - [6] Xie, Tianbao, et al. "Osworld: Benchmarking multimodal agents for open-ended tasks in real computer environments." arXiv preprint arXiv:2404.07972 (2024).
> > - [7] Rawles, Christopher, et al. "AndroidWorld: A dynamic benchmarking environment for autonomous agents." arXiv preprint arXiv:2405.14573 (2024).
> > - [8] Yin, Pengcheng, et al. "Natural Language to Code Generation in Interactive Data Science Notebooks." Proceedings of the 61st Annual Meeting of the Association for Computational Linguistics (Volume 1: Long Papers). 2023.
> > - [9] Li, Hongxin, et al. "SheetCopilot: Bringing software productivity to the next level through large language models." Advances in Neural Information Processing Systems 36 (2024).
> > - [10] Deng, Xiang, et al. "Mind2web: Towards a generalist agent for the web." Advances in Neural Information Processing Systems 36 (2024).
> > - [11] Huang, Qian, et al. "MLAgentBench: Evaluating Language Agents on Machine Learning Experimentation." Forty-first International Conference on Machine Learning. 2024.
> > - [12] Groq, Inc. (2024). Groq is fast AI inference. Retrieved from https://wow.groq.com/.
> > - [13] Yang, Jianwei, et al. "Set-of-mark prompting unleashes extraordinary visual grounding in gpt-4v." arXiv preprint arXiv:2310.11441 (2023).
> > - [14] Noah Shinn, Federico Cassano, Ashwin Gopinath, Karthik Narasimhan, and Shunyu Yao. Reflexion: Language agents with verbal reinforcement learning. Advances in Neural Information Processing Systems, 36, 2024.
> > - [15] Drouin, Alexandre, et al. "WorkArena: How Capable are Web Agents at Solving Common Knowledge Work Tasks?." ICLR 2024 Workshop on Large Language Model (LLM) Agents.
> > - [16] Koh, Jing Yu, et al. "VisualWebArena: Evaluating Multimodal Agents on Realistic Visual Web Tasks." ICLR 2024 Workshop on Large Language Model (LLM) Agents.
> > - [17] NeurIPS. (n.d.). NeurIPS Paper Checklist Guidelines. NeurIPS. Retrieved August 11, 2024, from https://neurips.cc/public/guides/PaperChecklist

---

> > > ### Comment · Reviewer_Lwbw · 2024-08-26
> > >
> > > I want to thank the authors for their detailed rebuttal. My concerns have been addressed therefore I am updating my rating from 3 to 6.

---

### Official Review · Reviewer_v3tL · 2024-07-25
**Good work with nontrivial contribution**

**Rating:** 7
**Confidence:** 3
**Clarity:** This paper is relatively easy to follow.

**Review:**

This paper is very well-motivated: as the LLM-based agent space is crowded with shiny demos, we need more specifically-designed benchmarks to evaluate their performance in domain-specific tasks. The authors have obviously put tremendously amount of efforts into this work.

Pros:
- Design of tasks are considerate: the proposed benchmark encompass both GUI and CLI interfaces
- Evaluated most popular closed-source/open-weight models
- Use real PC interactive environment (like a human would)

Minor cons:
-  Fixed prompt templates across models might have hindered model performances.
-  Complexity of the setup might make replication/adoption difficult

Overall, to the best of my knowledge, this work is a novel and I think it is a solid contribution to the broader community. Thus I recommend acceptance

**Strengths:**

This paper is very well-motivated: as the LLM-based agent space is crowded with shiny demos, we need more specifically-designed benchmarks to evaluate their performance in domain-specific tasks. I believe this work can be valuable to future applied/methodology research for automating data workflows.

**Additional Feedback:**

I have read the author's response and I am satisfied with the rebuttal. I will keep my decision of acceptance.

**Correctness:**

To my eyes, the data collection process is correct and I believe the experiment setup is also correct.

**Documentation:**

There are more detailed descriptions in supplementary material.

**Limitations:**

I think the authors have properly addressed the limitation.

**Opportunities For Improvement:**

For the main evaluation, the open-weight models lags far behind the closed-source ones. I am wondering have the authors use any model-specific instruction/chat tokens for the open-weight models? It could be beneficial to the community if the authors have specified that.

**Relation To Prior Work:**

To my knowledge, this paper seem to have sufficiently addressed related work.

**Summary And Contributions:**

This paper introduces Spider2-V, a new benchmark for evaluating multimodal AI agents on data science and engineering tasks. Spider2-V contains 494 real-world tasks and involve 20 professional software applications. The tasks are set in a real computer interaction environment. The authors provide setup configurations and evaluation metrics for each task. Experiments show state-of-the-art multimodal models fails to perform consistently across the landscape.

---

> ### Author Rebuttal · Authors · 2024-08-20
>
> We thank the reviewer for their time and attention to our work.
>
> **Q1: Open-sourced LLMs lag far behind closed-source models. Do you use model-specific instruction/chat tokens for open-weight models?**
>
> **A1: Yes, we use.** Sorry for the ambiguity about the prompt usage of open-weight models. The complete prompt we used can be splitted into 3 parts (Appendix E):
>
> **a) system message:** introduction to the environment, description on the action and observation space
>
> **b) user message (or task prompt):** task instruction, retrieved document context, observations from previous (determined by window size) and current interaction turns
>
> **c) assistant message:** exactly the LLM response, which includes history output actions and the action to be predicted
>
> When calling deployed open-sourced LLMs, we follow their official templates and insert messages above into the corresponding placeholders. For Llama-3-70B[1], the template is:
> ```txt
> <|begin_of_text|><|start_header_id|>system<|end_header_id|>
> {{here we insert system message}}<|eot_id|>
> <|start_header_id|>user<|end_header_id|>
> {{here we insert user message}}<|eot_id|>
> <|start_header_id|>assistant<|end_header_id|>
> ```
> The generation will terminate once a special `<|eot_id|>` token is emitted.
>
> As for the huge gap between closed- and open-source LLMs, we hypothesize the potential reasons are:
>
> **a) Intrinsic model capabilities:** Closed-source LLMs are more competent inherently than open-sourced models due to more sufficient pre-training, more investment on data of high quality and other unknown private techniques. This is a general consensus.
>
> **b) Context window size:** The open-sourced LLMs merely support context length less than $32k$, while closed-source LLMs at the top tier (e.g., GPT-4o, Gemini-Pro-1.5, Claude-3-Opus) all support much longer context ($\ge 128k$). Indeed, for main experiments in Table 3, the history window size is $3$ for closed-source LLMs, while $2$ for open-sourced ones since they cannot afford longer. And longer history window size always leads to better performances (proved in Figure 7).
>
> **c) Handling multi-modal observations:** The $3$ top-ranked LLMs all accept multi-modal input, thus we can provide the aligned screenshot and accessibility tree (SoM) as the observation. However, the chosen open-sourced LLMs can only handle text input (a11tree). This is another potential reason.
>
> Personally, we guess reason a) is still the dominant factor in task generalization. To bridge the performance gap between closed- and open-sourced models in real-world agent-environment interaction, agent tuning[2-4] is a promising direction (collecting agent trajectories and post-training on them for open-sourced LLMs).
>
> ----
>
> **Q2: Complexity of the setup might make replication/adoption difficult.**
>
> **A2:** We split the setup into two parts: a) benchmark-level, and b) instance-level.
>
> **1) Benchmark-level setup:** To use our benchmark from scratch, there are $4$ steps:
>
> 1. **Clone** the Github repo and create a `conda` environment.
> 2. **Install** software VMware Workstation Pro, and download our VMware snapshots (similar to Docker images) from `Huggingface`. Indeed, the download will be automatically done if not found locally when importing our environment class.
> 3. [Optional] **Register** authentic user accounts, and fill credentials into `.json` template files, if users want to evaluate on tasks (34%) involving relevant applications (e.g., BigQuery, Snowflake, etc. In total, 6 software). This works similar to providing an OpenAI API key. A simple glance over the Snowflake `.json` template:
> ```json
> {
>     "account": "https://{xxxxxxxx}.snowflakecomputing.com",
>     "user": "USER_NAME",
>     "password": "YOUR_PASSWORD"
> }
> ```
> Users can choose to test their agent on the task subset w/o authentic user accounts (66%).
>
> 4. **Run** our quick-start program which functions similar to OpenAI `gym`[5].
> ```python
> from desktop_env.envs.desktop_env import DesktopEnv
> # feel free to change the example
> example_path = 'examples/dagster/22ef9058-6188-422a-9c12-e6934e4ed936.json'
> with open(example_path, 'r') as infile:
> example = json.load(infile)
>
> env = DesktopEnv(action_space="pyautogui")
> obs = env.reset(task_config=example)
> obs, reward, done, info = env.step("pyautogui.rightClick()")
> input('Finish the task in the virtual machine manually and Press ENTER to evaluate ...')
> score = env.evaluate()
> print(f'Evaluation score: {float(score):.1f}')
> env.close()
> ```
>
> **2) Instance-level setup:** For each individual task, the setup (`env.reset()`) and evaluation (`env.evaluate()`) methods will be automatically conducted by our environment class after the `.json` metadata is loaded (see the quick-start program above). Benchmark users do not need to worry about task-specific setup.
>
> Unfortunately, to support the real-time interactive computer environment and enterprise-level professional applications, we are unable to bypass these odds and ends. However, **we are still in progress in improving user experience and simplifying deployment difficulty, such as supporting more platforms apart from VMware (e.g., virtual box, AWS, Google Cloud Project, etc.) and accelerating the agent-environment interaction efficiency**.
>
> ----
>
> **References:**
>
> - [1] Meta AI. Introducing meta Llama 3: The most capable openly available LLM to date, April 347 2024. URL https://ai.meta.com/blog/meta-llama-3/. Accessed: 2024-04-18.
> - [2] Chen, Baian, et al. "Fireact: Toward language agent fine-tuning." arXiv preprint arXiv:2310.05915 (2023).
> - [3] Chen, Zehui, et al. "Agent-FLAN: Designing Data and Methods of Effective Agent Tuning for Large Language Models." arXiv preprint arXiv:2403.12881 (2024).
> - [4] Xu, Yiheng, et al. "Lemur: Harmonizing Natural Language and Code for Language Agents." The Twelfth International Conference on Learning Representations.
> - [5] Brockman, Greg, et al. "Openai gym." arXiv preprint arXiv:1606.01540 (2016).

---

### Official Review · Reviewer_ULRM · 2024-07-25
**Likely an impactful dataset paper that may attract interest from the community investigating agent-based planning.**

**Rating:** 8
**Confidence:** 4

**Review:**

This paper is a great enhancement over the original Spider NL2SQL dataset, now targeting full Data Science workflows. The quality of the construction of the dataset is clearly high, with the involvement of 10 annotators for an average of 4h per task. The sourcing of tasks is also thoughtful, as well as the inclusion of 20 real tools used in the industry. 151 evaluation scripts have been custom developed to cover this variety of tasks, although softer metrics appear to be an opportunity for improvement (see Opportunities for Improvement). Overall, this is clearly a sound dataset paper that can be expected to attract substantial interest from the community investigating agent-based planning.

**Strengths:**

1. The dataset is well-grounded in real tasks and tools, as well as on human feedback through the evaluation scripts.
2. Tasks are executable in human-designed environments and include more than one format for their action spaces.
3. The paper includes multiple of the most competitive base models making use of different action space formats.

**Additional Feedback:**

No additional feedback.

**Clarity:**

For clarity, authors should specify "max number of steps" on line 75. The use of the word "tricks" on Table 5's caption and on line 246 should be replaced for more scientific language (e.g., "methods").

**Correctness:**

For the reasons already described, I believe this submission has very high soundness.

**Documentation:**

The submission includes sufficient documentation.

**Ethics:**

No special ethical concerns.

**Limitations:**

See "Opportunities for Improvement."

**Opportunities For Improvement:**

The main opportunity for improvement, given the complexity of the tasks in this dataset and the poor performance of current SoTA models, would be the development of softer evaluation metrics that could guide the agents via "partial credit." In a sense, my concern is that only returning a success flag (lines 90-91) might not be sufficient signal/feedback for agent-based planning within this dataset. The fact that all results in Table 3 make use of execution feedback seem to support this assessment.

**Relation To Prior Work:**

"AppAgent: Multimodal Agents as Smartphone Users" (available at https://arxiv.org/abs/2312.13771).

**Summary And Contributions:**

This submission introduces Spider2-V, a benchmark for Data Science workflows spanning 494 real-world tasks that make use of up to 20 industry tools over both GUI and CLI interfaces, with executable environments capable of verifying fully correct executions. The tasks are well distributed w.r.t number of action steps, their action space includes different representation formats (e.g., screenshots and accessibility trees), and multiple of the most competitive base models making use of said representations have been tested (i.e., GPT-4o, GPT-4V, Gemini, Claude, Qwen-Max, Llama3-70B, and Mixtral-8x7B) to show that this is indeed a very challenging benchmark for the current SoTA models.

---

> ### Author Rebuttal · Authors · 2024-08-20
>
> We thank the reviewer for their time and attention to our work.
>
> **Q1: Partial credit instead of sparse success flag for more fine-grained evaluation.**
>
> **A1:** Very interesting and insightful advice. During the construction of Spider2-V, we have ever considered evaluation with finer granularities, that is **setting milestones for different tasks** (e.g., obtaining a partial score 0.3 if the agent reaches a specific point). However, we abandoned this exciting idea in the current release for the reasons below:
>
> **(1) Fairness and objectivity of evaluation on the entire benchmark:** Let’s consider the following two CLI+GUI tasks:
>
> - **TASK I:** it requires 2 CLI commands followed by 2 GUI actions;
> - **TASK II:** it includes 1 CLI command followed by 2 GUI actions;
>
> Here are two agents A and B we want to compare, where
>
> - agent A succeeds to perform the CLI parts of both tasks (in total, 2+1 actions);
> - agent B succeeds to conduct the first CLI action in TASK I, and the first two actions in TASK II (in total, 1+2 actions);
>
> Now, let’s consider the following 2 evaluation metrics:
>
> **i) Setting milestones for each individual action:** If we treat each action as a scoring point in each task, agent A obtains (2/4+1/3)/2=0.42 score, while agent B attains (1/4+2/3)/2=0.46 score. We conclude that ==> **agent A < agent B**
>
> **ii) Setting milestones considering action complexity:** If we consider the semantics and intrinsic complexity of different actions (e.g., single code writing is equivalent to clicking multiple buttons in TASK II), and separate the CLI and GUI parts as two scoring points. Assume each part (CLI/GUI) has a 0.5 score. In this case, agent A gets (0.5+0.5)/2=0.5 score, while agent B gets (0+0.5)/2=0.25 score. We conclude that ==> **agent A > agent B**
>
> Evidently, **different methods of setting milestones lead to different conclusions**. Furthermore, in Spider2-V, CLI and GUI actions may even interleave with each other. And the intervals of these milestones (or how to assign fine-grained scoring points) differ across tasks due to the diversity and heterogeneity. This is complicated and sometimes **subjective**, which may lead to inconsistent and biased evaluation on the entire dataset.
>
> **(2) No oracle trajectory:** Another reason which prevents us from designing milestones is that during the real-time interaction with our dynamic environment, there is no oracle action trajectory. Our environment does not restrict agents on how to complete the task. Setting partially-finished milestones potentially **gives us erroneous evaluation if the agent gets around this scoring point and manages to finish the partial job in its own way**.
>
> In summary, we **utilize outcome-oriented automatic evaluation (that is, success rate) without checking intermediate results** to provide an unbiased, fair, and objective evaluation, and encourage agent self-exploration in the environment. BTW, if the task instruction explicitly stipulates the usage of certain tools, we will delve into specific implementation details in the evaluation script to ensure a thorough and accurate assessment.
>
> However, it is worth mentioning that using fine-grained scoring milestones is still a promising direction, especially when human evaluation (or LLM as user simulator[1][2]) is taken into consideration. We also hold a wait-and-see and open attitude towards this research direction.
>
> ----
>
> **Q2: What’s the maximum number of steps?**
>
> **A2:** Sorry for this missing important detail, we have added it in the updated version. Throughout our experiments, we set the “_maximum number of steps_” (more accurately, “interaction turns”) to be 15 for all agent baselines. Notice that although some tasks require more than 15 actions, we encourage the agent to predict multiple continuous actions in one response in the prompt for the sake of budget. This threshold can ideally cover all tasks.
>
> ----
>
> **Q3: The usage of the word “tricks” should be replaced.**
>
> **A3:** We will replace the casual usage with the formal and academic word “_methods_” in the caption of Table 5 and line 246.
>
> ----
>
> **Q4: Missing reference “_AppAgent: Multimodal Agents as Smartphone Users_”.**
>
> **A4:** We have cited this excellent work with the numeric label "[41]" in the Related Work section. AppAgent is a pioneer in leveraging LLM agents to complete real-world mobile tasks. Due to page limit, we only include “AndroidWorld” as a representative in the comparison Table 2. But we will also include “AppAgent” in the updated version.
>
> | Benchmark | Field | Exec. Env? | Ent. Serv.? | GUI Support? | # Apps / Sites | #  Exec.-based Eval. Func. | # Tasks |
> | ---- | ---- | ---- | ---- | ---- | ---- | ---- | ---- |
> | … |  | |  | | | | |
> | AppAgent[3]  | Android | $\checkmark$ | $\times$ | $\checkmark$ | 10 | 0 | 50 |
> | … |  | |  | | | | |
> | Spider2-V      | Data Science & Engineering w/ Computer Control | $\checkmark$ | $\checkmark$ | $\checkmark$ | 20 | 151 | 494 |
>
> ----
>
> **References:**
>
> - [1] Hu, Zhiyuan, et al. “Unlocking the potential of user feedback: Leveraging large language model as user simulators to enhance dialogue system.” Proceedings of the 32nd ACM International Conference on Information and Knowledge Management. 2023.
> - [2] Guan, Lin, et al. “Leveraging pre-trained large language models to construct and utilize world models for model-based task planning.” Advances in Neural Information Processing Systems 36 (2023): 79081-79094.
> - [3] Yang Z, Liu J, Han Y, et al. “Appagent: Multimodal agents as smartphone users.” arXiv preprint arXiv:2312.13771, 2023.

---

### Official Review · Reviewer_HHQN · 2024-07-27
**A challenging VLM evaluation suite using real-world programs and tasks**

**Rating:** 8
**Confidence:** 4
**Clarity:** The paper is well written and easy to…

**Review:**

The paper is well written and easy to follow. The evaluate is solid and the task construction is communicated effectively. The benchmark is non-trivial, with the strongest methods achieving only a 14% success rate. The usage of 20 real-world tools and a live running compute environment adds to the complexity of the problem while moving it closer to relevant real-world scenarios. The benchmark expands on prior work by using additional tools, a live compute environment, and tools that require internet access and real accounts to work with.

-------
REBUTTAL UPDATE
-------

The authors have addressed my concerns in the rebuttal, and thus I have adjusted my score from a 7 to a 8.

**Strengths:**

See Review Above.

**Additional Feedback:**

"developed 170 automatic task setup configurations and 151 customized evaluation metrics for each task.": 151 metrics for each task? Is this the correct wording, or did the authors mean 151 evaluation metrics across all tasks?

While the benchmark is indeed tough and the performance of the methods have low success rate, the authors could benefit from reducing the repeated usage of terms such as "extremely", "terribly", "exceedingly", etc. Instead, it would be better to provide the raw numbers in their place.

How do the authors plan to ease reproducibility when the tool usage requires a real-world account? Is it expected that those reproducing the work would pay for accounts on each of the tools to get access to them?

**Correctness:**

The paper appears correct, and primarily claims that current LLMs and VLMs are not sufficiently strong to handle complex real-world tasks consistently, which aligns with current expectations.

**Documentation:**

Yes, there is sufficient details on the benchmark/dataset suite.

**Ethics:**

I don't see any ethical concerns.

**Limitations:**

Limitations have been adequately addressed.

**Opportunities For Improvement:**

If step count is the biggest indicator of task difficulty, then it would be useful to plot the success rate for each value of step count for each method.

Runtime / compute complexity of the benchmark, both in terms of the LLMs/VLMs and the live compute environment, are important to document and mention in the paper.

**Relation To Prior Work:**

The authors clearly list prior work in Table 2.

**Summary And Contributions:**

The paper introduces Spider2-V, a challenging suite of 494 tasks using 20 real-world programs for evaluation of LLM/VLM models. Even the strongest VLM, GPT4V, achieves only a 14% success rate. On the hardest tasks with >15 steps, GPT4V achieves a 1.2% task success rate.

---

> ### Author Rebuttal · Authors · 2024-08-20
>
> We thank the reviewer for their time and attention to our work.
>
> **Q1: Success rate for each value of step count.**
>
> **A1:** Good Point. Here's a more detailed table which gives concrete success rates, and we will add the corresponding line chart in the updated version.
>
> | Step Count | 2      | 3      | 4      | 5      | 6      | 7      | 8     | 9      | 10   | 11   | 12   | > 12 |
> | ---- | ---- | ---- | ---- | ---- | ---- | ---- | ---- | ---- | ---- | ---- | ---- | ---- |
> | GPT-4o | 35.7 | 50.0 | 50.0 | 31.3 | 33.3 | 15.4 | 9.6  | 12.5 | 2.5  | 5.6  | 5.6 | 0.7   |
> | GPT-4V | 64.3 | 31.3 | 50.0 | 22.9 | 20.8 | 15.4 | 15.4 | 7.5 | 10.0 | 2.8 | 5.6 | 2.1   |
>
> In general, with the increase of action step numbers, the performance decreases stably.
>
> ----
>
> **Q2: Runtime / Compute complexity of the benchmark.**
>
> **A2:** We will supplement these significant statistics.
>
> **1) Average running time and cost per task:** Note that the running time complexity is tightly connected to the maximum interaction turns, trajectory window size and network delay when calling APIs.
>
> | Model | Expected Running Time | Average Prompt Tokens | Average Completion Tokens | Average Cost Per Task |
> | ----------- | ----------------- | --------------- | ------------ | ------------ |
> | GPT-4o  | 8.7 minutes | 0.32M        | 1.6K        | $1.64      |
>
> On average, each task costs $1.64 with GPT-4o. The detailed configuration is (also the most powerful agent in our main experiment):
> - action_space = pyautogui code
> - observation_space = SoM
> - prompt tokens include image pixels
> - max_turn = 15
> - trajectory_window_size = 3
> - w/ retrieval augmentation (top_k = 4, chunk_size = 512 tokens)
>
> **2) Real-time environment delay:** The Expected Running Time for each task can be further decoupled into:
> 1. environment setup time;
> 2. response time when calling APIs or querying LLMs (may depend on the network traffic);
> 3. time delay to perform each action in the computer desktop (this is controllable, and we set the time delay to 1s for the action to take effect);
> 4. time delay to obtain the screenshot and accessibility tree (observations) after each action is grounded;
> 5. evaluation time at the end of the episode.
>
> | Environment Setup  | Per Action Time Delay | Per Observation Time Delay | Evaluation Time |
> | ----------- | ----------- | ----------- | ----------- |
> | ~ 1 minute   | 1 second    | ~ 2 seconds        | < 1 minute          |
>
> Accordingly, we can find that the majority time is spent on multi-turn API calls (calculated by elimination) due to long prompt tokens and slow inference time. Interaction efficiency is a promising future direction.
>
> ----
>
> **Q3: Counting of environment setup methods and evaluation metrics.**
>
> **A3:** Thanks for pointing out this typo. The statistics (e.g., 170/151) denote different environment setup and evaluation methods across the entire benchmark, not for a single task. **We will delete the phrase “for each task” in line 56.**
>
> ----
>
> **Q4: Avoid using ambiguous words such as “extremely”, “exceedingly”.**
>
> **A4:** Thanks for your advice. We will replace these unclear adjectives with concrete numbers to reinforce the reader's impression of the existing model's capabilities. Concretely,
> - line 201: For those open-source LLMs, the success rate is less than 2%, approaching insignificance.
> - line 205: The success rates for data warehousing and traditional data processing tasks are both less than 10%.
> - line 220: And for those tasks with step count larger than 15, …
>
> ----
>
> **Q5: Ease reproducibility for tasks requiring real user accounts.**
>
> **A5:** Very good question. One hurdle of utilizing professional enterprise software is authentic user accounts.
>
> **a) w/o authentic user accounts:** In Spider2-V, more than half of the tasks (66%) can be launched in localhost. That is, we can start the Web server through Docker containers and leverage the default login account. Benchmark users do not need to register a new one and the task can be completed totally in localhost. You can choose this dataset split merely for easier reproduction.
>
> **b) w/ authentic user accounts:** For the other 34% tasks, we will provide a `.json` template file for each professional application which requires a real-world user account (e.g., BigQuery, ServiceNow, etc.). If benchmark users want to evaluate on them, to ensure successful automatic setup and evaluation, they need to register each account manually and fill personal credentials into different `.json` templates (in total, 6 templates/software across Spider2-V). Here is a simple glance over the Snowflake template:
> ```json
> {
>     "account": "https://{xxxxxxxx}.snowflakecomputing.com",
>     "user": "USER_NAME",
>     "password": "YOUR_PASSWORD"
> }
> ```
>
> Once the credentials are filled, the automatic environment setup and customized evaluation for each task will perform the remaining work. This works similar to setting the OpenAI API key (previous work WorkArena[1] also functions in this way). We also provide a detailed step-by-step tutorial in the Github repository on how to obtain each json field for each software.
>
> **With respect to credit card payment, we assure you that all these tasks do not need additional payments throughout the entire process.** Because we only exploit the Sandbox[2] or Trial[3] accounts, which means the free tier can adequately cover all consumption over the entire benchmark. You even do not need to add your credit card when signing up.
>
> ----
>
> **References:**
>
> - [1] Drouin A, Gasse M, Caccia M, et al. “WorkArena: How Capable are Web Agents at Solving Common Knowledge Work Tasks?” ICLR 2024 Workshop on Large Language Model (LLM) Agents.
> - [2] Google Support. “Use the BigQuery sandbox.” Google Docs. Retrieved August 11, 2024, from https://support.google.com/docs/answer/9703715?hl=en
> - [3] Snowflake Inc. (n.d.). “Managing trial accounts.” Snowflake Documentation. Retrieved August 11, 2024, from https://docs.snowflake.com/en/user-guide/admin-trial-account

---

> > ### Comment · Reviewer_HHQN · 2024-08-20
> > **Response to Authors**
> >
> > Thank you for the detailed rebuttal. The response has addressed my remaining concerns, and I have adjusted my score accordingly.

---

### Author Rebuttal · Authors · 2024-08-20

We would like to thank all reviewers for their patience and precious time, as well as many constructive and  insightful suggestions which help us further polish our work. We are also delighted that our benchmark is considered “_non-trivial_” (reviewer HHQN), “_Top 50% of accepted papers_” and “_attract substantial interest from the community_” (reviewer ULRM), “_solid contribution_” (reviewer v3tL) and “_interesting in context of workflow automation agents_” (reviewer Lwbw).

We have addressed specific queries in each individual response. The key updates are summarized as follows:

- **Missing implementation details and experiments.** We have added:
    - runtime complexity and cost (see Q2 for reviewer HHQN and Lwbw);
    - analysis on different action step numbers (see Q1 for reviewer HHQN);
    - prompt usage for open-sourced models (see Q1 for reviewer v3tL);
    - separate limitation and social impact sections (see Q1 for reviewer Lwbw);
    - performance comparison with other benchmarks (see Q4 for reviewer Lwbw).
- **How to ease reproducibility regarding tasks involving authentic user accounts.** The general steps is demonstrated in Q5 for reviewer HHQN and Q2 for reviewer v3tL.
- **Use partial credit instead of sparse success rate for evaluation.** This interesting topic is discussed in Q1 for reviewer ULRM.
- **Clarification on the contribution and novelty of Spider2-V.** The main contributions and comparison to previous benchmarks are highlighted in Q6 for reviewer Lwbw.
- **Other suggestions, typos, word misuse, and missing reference.** Detailed issues are addressed for each reviewer.

Once again, we sincerely thank all reviewers for their valuable and professional contributions towards enhancing our manuscript. We believe these revisions have significantly strengthened our paper. If there's any need for further clarification to help finalize the assessment of our work, please don't hesitate to let us know.

Thank you for your review!

---

### Decision · Program_Chairs · 2024-09-26

**Decision:**

Accept (Spotlight)

**Comment:**

This works provides a challenging new benchmark for the evaluation of VLMs on full real-world data engineering assembly/programming problems combining both visual, interactive and textual challenges . Even the largest currently available models find the defined tasks difficult. The benchmark is well documented and provides virtual machines to actual perform the tasks. It is highly accessible for the community. Overall, a great example for the track.